# REAL-VAS: A REALWORLD VIDEO AMODAL SEGMENTATION DATASET

## ABSTRACT

Amodal video object segmentation is fundamentally limited by the absence of datasets that combine real-world complexity with precise ground-truth annotations. To address this, we present Real Video Amodal Segmentation (Real-VAS), a new large-scale, zero-shot evaluation dataset. We introduce a novel data generation pipeline that enables the creation of pixel-perfect amodal ground truth for real-world video without relying on human estimation or expensive 3D reconstruction. Our dataset is structured into two challenging scenarios: dynamic Occlusion, created by compositing two clips of moving objects, and a unique Container category featuring complex, physically constrained interactions. These container scenarios–On Surface Containment, Articulated Containment, and Mobile Containment–allow us to generate precise ground truth by simulating an object's motion based on its container's tracked transformation. As a result, Real-VAS provides a diverse and challenging benchmark for evaluating amodal segmentation models on realistic video with the precision of synthetic data. The dataset and our generation code will be made publicly available.

## 1 INTRODUCTION

Video Object Segmentation (VOS) is the problem of densely tracking the visible pixels of an object. Methods such as SAM2 (Ravi et al., 2024) and Mask2Former (Cheng et al., 2022) have made remarkable progress on this problem in recent years, offering very high accuracy and robustness. However, in humans our visual system has a strong notion of *object permanence* (An et al., 2022) – the idea that objects maintain their identity and continuity over time, regardless of visibility – allowing us to predict where an object exists and how it may have deformed, even when occluded. This concept is called *amodal completion* (Ao et al., 2023; Nanay, 2018) and is the focus of this paper. Robust amodal completion is critical for many real-world applications, from autonomous driving, where vehicles must track occluded pedestrians, to robotic manipulation, where a robot needs to infer an object's full shape to grasp it in a cluttered environment.

Progress on machine Video Amodal Segmentation (VAS), i.e. densely tracking object segmentations *through* occlusions, has received relatively less attention and progress has not been as rapid (Zhu et al., 2019; Kortylewski et al., 2020). One of the reasons for this limited progress is the lack of sufficiently challenging datasets with ground truth segmentations behind occlusions. Existing datasets suffer from several important weaknesses. On the one hand, synthetic datasets provide exact ground truth but are limited in realism and diversity, particularly for dynamic objects. On the other hand, obtaining ground truth segmentations for objects behind occlusions in real-world video is an extremely difficult and resource intensive task (Van Hoorick et al., 2023). Possible solutions include multi-camera or 3D setups to record ground truth, which is not scalable, or human best-guess annotations behind occlusions, which is not accurate. The real-world benchmarks that are available do not provide accurate (or any) ground truth segmentation behind occlusions. Instead, they either provide ground truth only on unoccluded frames (in which case the benchmark is really testing reidentification (Zheng et al., 2016; Bergmann et al., 2019; Zhang et al., 2022; 2021) rather than amodal segmentation) or rely on hand-drawn, approximate masks. Finally, existing datasets do not contain sufficiently challenging scenarios such as occlusion by a container (which requires reasoning about the motion of the container) or objects never re-appearing after occlusion (which requires extrapolation without relying on reidentification).

Figure 1: An overview of our Real-VAS data generation pipeline. Occlusion pipeline: from static camera videos, we automatically find compatible pairs of clips that can be composited to form realistic occlusions. Occluder objects and their effect (e.g. shadows) are composited over occludee objects for which accurate ground truth is computed from the unoccluded sequence. Containment pipeline: we identify three scenarios in which tracking the container is sufficient to provide ground truth tracking of the contained object. For both pipelines, additional camera motion can be subsequently emulated by cropping to a narrower moving field of view.

In this paper we propose scalable methods for creating a largescale VAS dataset using real-world videos and without specialist acquisition setups. We propose two pipelines (see Fig. 1). The first allows arbitrary rigid or non-rigid, possibly dynamic occluders to be photorealistically composited over arbitrary occludee objects. The second allows complex occludee-container relationships to be accurately tracked. We use these methods to create the most challenging VAS dataset to date, Real-VAS, which will be made publicly available along with code for our dataset generation pipeline.

## 2 RELATED WORK

**Amodal Segmentation Methods** Current amodal segmentation methods can be categorized by architectural paradigm and temporal context usage. Architecturally, feedforward models like TCOW (Van Hoorick et al., 2023) directly predict amodal states, while generative approaches treat completion as conditional in-painting using diffusion priors. Image-based diffusion methods (Ozguroglu et al., 2024; Zhan et al., 2024b) process frames independently, whereas video-based approaches (Hudson & Smith, 2024; Chen et al., 2025; Lee et al., 2025) operate offline on sequences for temporal consistency. These methods rely on various contextual cues, often requiring additional inputs like modal masks (Ozguroglu et al., 2024; Zhan et al., 2024b; Hudson & Smith, 2024; Chen et al., 2025; Lee et al., 2025) or bounding boxes (Zhan et al., 2024b), and employ techniques like depth estimation (Ozguroglu et al., 2024; Hudson & Smith, 2024; Chen et al., 2025) or target region estimation (Hudson & Smith, 2024).

| Metric | Real Datasets | | | | Synthetic Datasets | | | |
|---|---|---|---|---|---|---|---|---|
| | **Real-VAS** Ours | **COCOA** Zhu et al. (2017) | **COCOA-cls** Follmann et al. (2019) | **D2S** Follmann et al. (2018) | **MOVi-MC-AC** Moore et al. (2025) | **SAIL-VOS 3D** Hu et al. (2021) | **SAIL-VOS** Hu et al. (2019b) | **DYCE** Ehsani et al. (2018) |
| Image or Video | Video | Image | Image | Image | Video | Video | Video | Image |
| Accurate Annotations | ✓ | ✗ | ✗ | ✗ | ✓ | ✓ | ✓ | ✓ |
| Video Scenes | 400 | - | - | - | 2,041 | 203 | 201 | - |
| Scene Images | 21,436 | 5,073 | 3,499 | 5,600 | 293,904 | 237,611 | 111,654 | 5,500 |
| Unique Objects | 160 | - | 80 | 60 | 1,033 | 178 | 162 | 79 |
| Occlusion Prevalence | 65.0% | 60.7% | 49.0% | 56.9% | 69.3% | - | 87.2% | 82.3% |
| Mean Occlusion Severity | 60.0% | 18.8% | 10.7% | 15.0% | 45.2% | - | 56.3% | 27.7% |

Table 1: Comparison of amodal segmentation datasets. Occlusion Prevalence measures how often each instance is occluded. Mean Occlusion Severity measures what proportion of an object is occluded on average when they are occluded.

**Amodal Segmentation Datasets** Existing datasets (Table 1) face a trade-off between annotation accuracy and real-world fidelity. Synthetic datasets achieve perfect ground truth via physics modeling (Van Hoorick et al., 2023; Hu et al., 2019a; Reddy et al., 2022; Fan et al., 2023; Mohan & Valada, 2022), motion tools (Tangemann et al., 2021; Yao et al., 2022; Gao et al., 2023; Girdhar & Ramanan, 2019; Shamsian et al., 2020), and image occlusion (Follmann et al., 2018; Ehsani et al., 2018; F. Dormann et al., 2007), but suffer from sim2real gaps (Höfer et al., 2021; Wechsler et al., 2024). Real-world approaches face annotation challenges since occluded regions require human estimation rather than ground truth measurement—a process that is time-intensive, inaccurate, and subjective (Zhu et al., 2017; Qi et al., 2019; Follmann et al., 2019). Large-scale benchmarks like TAO-Amodal (Hsieh et al., 2023) use human estimates of bounding boxes but not pixel-level masks. 3D acquisition methods (Zhan et al., 2024a; Li et al., 2023) provide precise masks but lack scalability.

Compositing approaches, successful in other vision tasks (Walawalkar et al., 2020; DeVries, 2017; Bochkovskiy et al., 2020; Zhao et al., 2020; Kini et al., 2024), often rely on simplistic overlays in amodal segmentation (Ozguroglu et al., 2024; Lee et al., 2025), creating realism gaps. Additionally, existing datasets focus on simple "pass-by" occlusions, neglecting challenging containment scenarios where objects physically interact (Van Hoorick et al., 2023). TCOW's real-world evaluation only annotates unoccluded frames, reducing the task to re-identification rather than true amodal segmentation (see Table 2).

|  | Rubric Office | Rubric Cup Games | Rubric DAV/YTB |
|---|---|---|---|
| TCOW | 69.4 | 38.3 | 52.8 |
| SAM2 | **72.7** | **55.8** | **68.9** |

Table 2: Comparison of TCOW (Van Hoorick et al., 2023) and SAM2 (Ravi et al., 2024) mean IoU metrics on amodal segmentation datasets. Despite SAM2 providing only *modal* segmentation, it outperforms TCOW's amodal result.

To address these significant gaps in existing work, we introduce Real-VAS, a novel dataset and data generation pipeline for zero-shot video amodal segmentation. Our approach is designed to synthesise the strengths of prior methods while mitigating their respective drawbacks. Our *Occlusion* methodology uses a compositing pipeline on real-world footage, achieving the pixel-perfect ground truth of synthetic data without the sim2real gap or the imprecision and high cost of manual annotation. To overcome the realism gap of prior compositing work, our Occlusion subset features dynamic interactions between moving objects and includes emulated camera motion. Furthermore, to move beyond simple pass-by events and benchmark physical reasoning, we introduce the novel *Containment* category with three physically-grounded scenarios—On Surface Containment, Articulated Containment, and Mobile Containment.

## 3 TASK

The *zero-shot video amodal segmentation* (ZS-VAS) task is defined as follows. We are given as input a $T$-frame video $\mathbf{x} \in \mathbb{R}^{T \times H \times W \times 3}$ along with a binary query mask $\mathbf{m}_q \in \{0, 1\}^{H \times W}$ which segments the object of interest in some reference frame (assumed to be frame 1). It is assumed that the object is completely unoccluded in the reference frame. The query mask may itself have been computed from a prompt of some form (e.g. text or point clicks). The goal is to learn a function $f$ that estimates amodal segmentation masks for the tracked object in every frame, $\mathbf{m}_o = f(\mathbf{x}, \mathbf{m}_q)$, where $\mathbf{m}_o \in \{0, 1\}^{T \times H \times W}$ is the binary object mask in every frame. This must delineate the object even behind occluders and even when none of the object is visible based on information from surrounding frames and the context of the object itself. This initialisation differs from datasets like TAO-Amodal (amodal object tracking) and SAIL-VOS (synthetic amodal object segmentation), where the object of interest may be occluded from the start, often requiring a model to rely on pre-existing class knowledge for identification. The zero-shot paradigm is therefore class-agnostic; it evaluates the more general and realistic ability of a model to segment any novel object based purely on the provided query, rather than its familiarity with specific object categories.

We created this dataset to be an evaluation task, providing a wide-ranging benchmark of real-world scenarios, dynamic object classes, and physically complex interactions designed to test the limits of zero-shot amodal segmentation and aid in the development of the field.

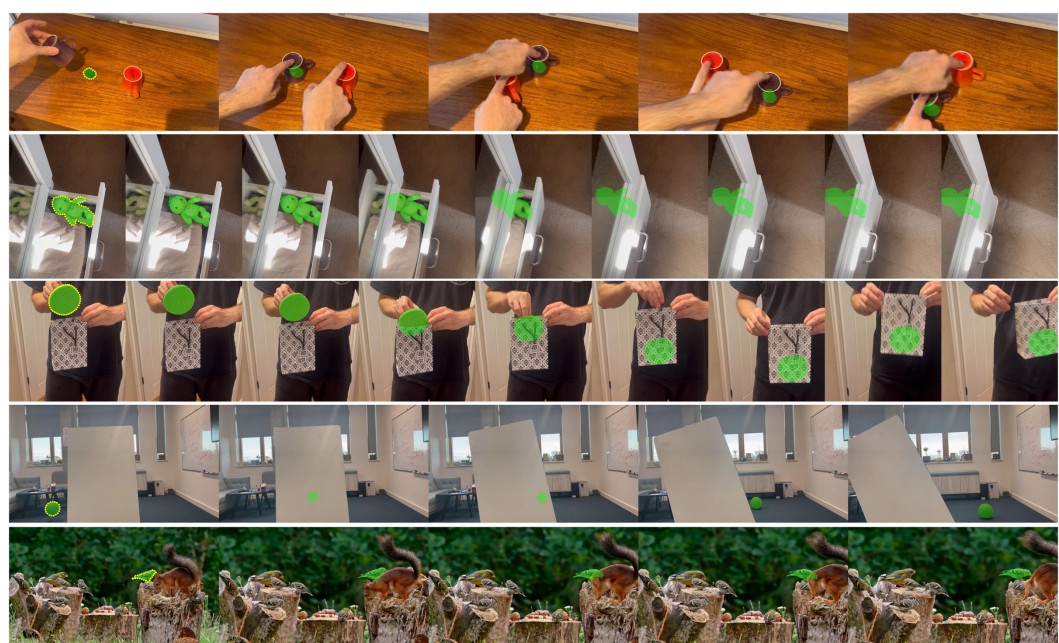

Figure 2: Qualitative examples from the Real-VAS dataset. The yellow dashed line indicates the initial query mask, followed by the ground-truth amodal segmentation frames. The rows showcase our scenarios: On Surface Containment, Articulated Containment, Mobile Containment, and two Dynamic Occlusion examples. One captured in-house (row 4) and sourced from the web (row 5).

## 3.1 EVALUATION METRICS

The standard metrics used for evaluating amodal segmentation are mean Intersection-over-Union (mIoU) and mIoU computed over only occluded pixels, referred to as mIoU$_{occ}$ (Chen et al., 2025; Gao et al., 2023; Fan et al., 2023). However, we argue that mIoU is a highly misleading metric for amodal segmentation evaluation as high scores on non-occluded frames can obscure poor performance on heavily occluded ones. This means that modal methods can seemingly outperform amodal methods (see Table 3). For this reason, we report mIoU$_{occ}$ and two additional metrics: mIoU$_{fo}$ is

|  | SAM2 | TABE | Diffusion VAS |
|---|---|---|---|
| mIoU | 0.621 | 0.542 | 0.655 |
| mIoU$_{occ}$ | 0.066 | 0.207 | 0.262 |

Table 3: Standard mIoU is a misleading metric for amodal segmentation. The table shows how a purely modal method (SAM2) can achieve a high overall score while completely failing on occluded regions (mIoU$_{occ}$), a critical flaw masked by the single, averaged metric.

mIoU computed over frames with any level of occlusion and mIoU$_{ffo}$ is a stricter version computed over *fully occluded* frames, where no object pixels are visible:

$$\text{mIoU}_{\text{fo}} = \frac{\sum_{i \in F_{\text{occ}}} |\hat{A}_i \cap A_i|}{\sum_{i \in F_{\text{occ}}} |\hat{A}_i \cup A_i|}, \quad \text{mIoU}_{\text{ffo}} = \frac{\sum_{i \in F_{\text{full}}} |\hat{A}_i \cap A_i|}{\sum_{i \in F_{\text{full}}} |\hat{A}_i \cup A_i|}, \quad \text{mIoU}_{\text{occ}} = \frac{\sum_{i \in F_{\text{occ}}} |(\hat{A}_i - M_i) \cap (A_i - M_i)|}{\sum_{i \in F_{\text{occ}}} |(\hat{A}_i - M_i) \cup (A_i - M_i)|},$$

where $M_i$ is the ground-truth **modal mask**, $\hat{A}_i$ the **predicted amodal mask**, and $A_i$ the **ground-truth amodal mask** for frame $i$. We define $F_{occ}$ as the set of **occluded frames** and $F_{\text{full}} \subseteq F_{\text{occ}}$ as the set of **fully occluded frames** (for which $M_i = \emptyset$).

## 4 REAL-VAS DATASET

We introduce Real-VAS, a novel dataset with precise ground-truth annotations for the evaluation of video amodal segmentation. As shown in Fig. 1, Real-VAS is constructed using two distinct pipelines, 'occlusion' and 'container', to test performance across a diverse range of amodal scenarios, while remaining automatic and scalable to enable the creation of largescale datasets. To ensure the final quality of the dataset, we include two manual quality control checks (Appendix A.5).

We use two sources of raw data: controlled scenes captured in-house and scraping diverse videos from public video repositories. This hybrid approach allows us to combine staged, highly controlled interaction scenarios with the vast diversity of real-world footage. Our in-house data was captured using a tripod-mounted iPhone 15, recording at a high frame rate (slow-motion) with locked focus and exposure to ensure consistent framing and minimal motion blur for dynamic objects. For the externally sourced data, we utilised footage from both existing large-scale video datasets (Shuai et al., 2022; Chavdarova et al., 2017) and in-the-wild videos from public platforms such as YouTube, filtered to provide videos from static cameras.

## 4.1 REAL-VAS OCCLUSION

We use a compositing strategy to obtain precise masks for occluded objects. For scenes we capture ourselves, we record two clips with a static camera: one showing the object to be occluded (the "occludee") without obstruction, and a second clip showing the "occluder". For scraped video data, we propose an automated pipeline for finding suitable pairs of sub-clips that contain a compatible occluder and occludee. We begin by feeding text prompts from a predefined list of dynamic object classes (see Appendix A.1) into Language Segment-Anything (Medeiros, 2025). This open-vocabulary model segments the corresponding objects by combining a grounding detector (Zhang et al., 2023) with SAM2. The resulting masks are then tracked across all frames with SAM2.

To ensure the dataset is both high-quality and challenging, a dynamic object checker filters for sequences where both the occluder and occludee are dynamic. To determine if an object is dynamic, we calculate a mean motion score $\mathcal{S}$ based on the trajectories of its points over time, gathered using CoTracker3 (Karaev et al., 2024). Let the set of trajectories for a single object be represented by $\mathbf{T} \in \mathbb{R}^{T \times P \times 2}$, where $T$ is the total number of frames (time steps), $P$ is the number of points tracked on the object, $\mathbf{p}_{t,i} \in \mathbb{R}^2$ denotes the $(x, y)$ coordinates of the $i$-th point at frame $t$. We define the *mean motion score*, $\mathcal{S}$, as the average displacement (Euclidean distance) of all points over all frames:

$$\mathcal{S} = \frac{1}{PT} \sum_{i=1}^{P} \sum_{t=1}^{T-1} \|\mathbf{p}_{t+1,i} - \mathbf{p}_{t,i}\|_2 \,.$$

An object is classified as **dynamic** if its mean motion score exceeds a predefined minimum motion threshold, $\mathcal{S} > \tau$. This formulation effectively captures the object's overall movement by averaging the speed of its constituent points across the entire video sequence.

Now that we have segmentation masks for dynamic objects in the video we can find where composited occlusion can be created. The core purpose of this method is to **discover and validate high-quality object occlusion events** by finding the best way to temporally align two separate video sequences of object masks ($M_1$ and $M_2$). The process is structured into two main stages: Candidate Generation and Rigorous Filtering.

### 4.1.1 CANDIDATE GENERATION VIA CROSS-CORRELATION

The algorithm first identifies the most promising temporal alignments by calculating an **overlap score** for every possible time shift ($\Delta t$) between the two valid mask sequences, i.e. a **cross-correlation**. The score $S(\Delta t)$ for a given shift is the sum of the spatial intersection (the overlapping pixel area) over the duration of the sequence:

$$S(\Delta t) = \sum_t |M_1[t] \cap M_2[t - \Delta t]|,$$

where $|\cdot|$ denotes the number of pixels in the intersection. We compute this score for all possible shifts and sort them, creating a ranked list of the best potential occlusion events.

### 4.1.2 RIGOROUS FILTERING PIPELINE

Next, the algorithm iterates through the top-ranked candidates and subjects each one to a comprehensive set of validation checks to ensure the resulting composite scene is physically plausible, interesting, and non-redundant. For each candidate, the objects are first assigned **foreground (FG)** and **background (BG)** roles based on their relative depth values. The sequence is then trimmed or rejected if it fails any of the following key tests:

**Physical plausibility:** We perform several automated checks using monocular depth estimates from Depth Anything V2 (Yang et al., 2024). First, we prevent *Reverse Occlusion* by analysing the pixel region where the foreground (FG) and background (BG) masks overlap. The sequence is rejected if any FG pixel in this region has a greater depth value than its corresponding BG pixel. Second, our *"On Top" Detection* filters out scenarios where objects appear to improperly occupy the same 3D space. Finally, we check for *External Occlusion* by detecting if an object's size unexpectedly diminishes; if the depth map indicates a closer surface at the point of shrinkage, we infer an external occlusion and discard the candidate.

**Quality & Interest Checks:** First, every sequence must have a clean start, meaning the background object is fully visible in the initial frame. Next, the interaction must result in sufficient occlusion, where the occluded area exceeds a minimum threshold.

**Redundancy Checks:** A similarity test discards candidate sequences that are too visually similar to previously accepted ones, while a frame overlap check prevents the same source frames from being used for both the foreground and background objects.

Only candidates that successfully pass this entire pipeline are considered valid and passed forward to the next stage of the pipeline.

**High Resolution Sequence Generation** To ensure computational efficiency, the preceding steps operated on video clips temporally downsampled to 1-3 fps. This frame rate provided sufficient temporal resolution to find candidate overlaps while minimising processing time. To convert the coarse, low-framerate mask sequences (1-3 fps) into high-quality masks at the video's native framerate, we employ a propagation-based approach once again using SAM2. This process involves an initialisation step followed by a focused, fine-grained segmentation pass. The process begins by identifying the first valid mask in the low-framerate sequence to serve as an initial prompt. However, temporal downsampling can miss the true first appearance of an object. To correct this, we perform a preliminary backwards process: the segmentation model is run on a reversed video at high framerate, immediately preceding this initial mask. This refinement step pinpoints the object's true starting frame and provides a more accurate initial mask for the main forward process. To generate fine-grained masks with high efficiency and precision, we first create a dense sequence of guiding bounding boxes. This is done by linearly interpolating the sparse set of boxes from the low-framerate sequence, one for every high-framerate frame, and adding a motion-compensation padding. The segmentation model is then applied exclusively within these tight, content-focused crops of the high-framerate video. This approach significantly boosts both processing speed and the resulting boundary detail. Finally, these high-detail mask segments are re-projected onto the full-frame canvas at their original coordinates, and the backwards process is run again to refine the initial frames of the new, high-resolution sequence.

**Compositing** With our high-resolution sequences, we use a multi-stage compositing pipeline to create the final dataset by realistically overlaying the foreground object, $I_{fg}$, onto the background scene, $I_{bg}$. We do so using a standard alpha blend with a soft alpha matte, $\alpha_{fg} \in [0, 1]$, to allow for seamless, semi-transparent boundaries. The composite image, $I_{comp}$, is given by:

$$I_{comp} = (1 - \alpha_{fg}) \cdot I_{bg} + \alpha_{fg} \cdot I_{fg}. \quad (1)$$

In practice, the alpha mattes are provided by Generative Omnimatte (Lee et al., 2025). This model captures both the foreground object and its secondary effects on the scene, such as cast shadows or reflections, and represents them in the soft alpha matte. Including these effects significantly increases the realism of the final composite, as shown in Fig. 3.

## 4.2 REAL-VAS CONTAINMENT

Our Container category is composed of three physically distinct scenarios (see Fig. 1). While the initial interactions are unique, all three are built upon a unified principle for generating precise ground truth: once an object is fully contained, its motion is dictated by the motion of its container. To enable this, we enforce a physical constraint by using form-fitting object-container pairs. Computationally, this process relies on two key models. First, SAM2, is used to track an object while it is visible, establishing its last known mask before occlusion. Second, once the object is hidden, a point tracker, CoTracker3, captures the container's precise transformation. This transformation is then directly applied to the object's last known mask, treating them as a single rigid unit. The difference between scenarios lies in how these models are applied during the initial interaction:

Figure 3: Comparison of data compositing with and without Generative Omnimatte. The bottom row, using our method, successfully captures secondary effects such as shadows and reflections. This results in a more realistic composite compared to the baseline method shown in the top row.

**On Surface Containment:** inspired by the shell game, the object is static. The CoTracker3 logic is applied to the container from the moment of full occlusion.

**Articulated Containment:** we first use SAM2 to track the object's visible movement as it enters a fixture (e.g., a drawer). Once occluded, we switch to using CoTracker3 on the container.

**Mobile Containment:** requires a multi-stage entry where we first reconstruct the partially occluded mask, simulate the object settling, and only then engage the CoTracker3 logic on the container.

## 4.3 EMULATED CAMERA MOTION

To ensure our dataset is as realistic and varied as possible, we emulate camera motion for scenes where the original data was collected with a static camera (Fig. 4). This is mainly applied for our **Real-VAS Occlusion** set, which requires static camera data as input, whereas our container logic allows for camera motion. Our approach applies a diverse set of predefined camera motion profiles (e.g., *pans, zooms, and translations*) to the static source videos.

To ensure that the emulated motion does not corrupt the data by pushing key objects out of frame, we employ an **automated validation and selection pipeline**. Each potential motion profile is used to generate a candidate sequence, which is then evaluated against two critical criteria. First, we ensure the *preservation of visibility*, verifying that the emulated motion does not push the primary objects (particularly the occludee) out of the frame. Second, we check for the *preservation of occlusion quality*, ensuring that the interaction is maintained and the degree of occlusion is not substantially reduced from the original static version. A suitable motion profile is then randomly selected from the pool of candidates that successfully pass these checks, introducing valuable diversity into our dataset. If no emulated motion satisfies our quality constraints for a given scene, the camera remains static, thereby preserving the integrity of the data.

## 4.4 DATASET STATISTICS

Table 1 presents a comparison between Real-VAS and other notable amodal segmentation datasets. Our contribution is significant in several key areas. First, Real-VAS provides precise annotations, removing the imprecision of human estimation found in other real-world datasets. Second, in terms of scale, it contains almost four times more frames than the next largest real-world benchmark. This scale is matched by its diversity, offering a substantially larger number of unique object classes. Finally, it is specifically designed for the amodal task, featuring both a higher occlusion prevalence and a much greater mean occlusion severity, making it a more challenging and suitable benchmark for evaluating amodal models. When compared to synthetic datasets, Real-VAS naturally trades sheer scale for real-world fidelity. While synthetic benchmarks like MOVi-MC-AC (Moore et al., 2025) offer a greater volume of scenes and classes, Real-VAS is intentionally curated to feature more challenging and realistic occlusions. This is evidenced by its mean occlusion severity, which is higher than that of most synthetic datasets. This focus on realism and the quality and difficulty of occlusions, rather than just the quantity, makes Real-VAS a suitable evaluation dataset for amodal segmentation. Further dataset metrics are present in Appendix B.2.

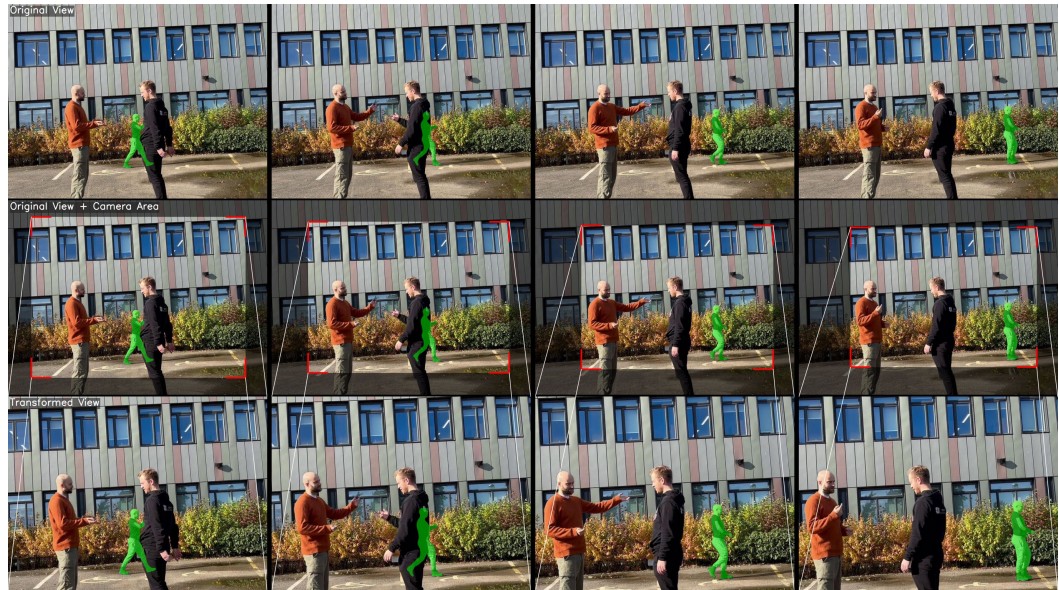

Figure 4: An example of a emulated camera motion. This figure demonstrates a dynamic zoom and pan by showing the original frame, the specific sub-region that constitutes the new camera view, and the resulting transformed frame.

While our comparison in Table 1 is comprehensive, we have deliberately excluded some related datasets due to fundamental differences in their annotation formats. For instance, TAO-Amodal (Hsieh et al., 2023), despite being a large-scale real-world benchmark, is omitted as it only provides amodal annotations in the form of bounding boxes, which are not comparable to the pixel-level segmentation masks that our work focuses on. Similarly, the 'Rubric' real-world evaluation set from TCOW (Van Hoorick et al., 2023) is excluded because it exclusively contains modal annotations (visible parts) of objects preceding and following an occlusion, and therefore lacks the necessary amodal ground truth to evaluate a model's performance during an occlusion event itself.

## 5 EVALUATION

To benchmark performance on Real-VAS, we evaluated a diverse set of state-of-the-art (SOTA) models, as shown in Table 4. These benchmarks include image-based models (pix2gestalt, SDAmodal), video diffusion approaches (TABE, Diffusion VAS, Generative Omnimatte), and video transformers (TCOW). We excluded methods that output 3D amodal shapes from our benchmarks, such as Amodal3R (Wu et al., 2025), as the lack of ground-truth camera intrinsics in our dataset prevents a fair evaluation of their projected 2D segmentation masks. Several of these frameworks require additional inputs beyond the initial query mask. For consistency, where a *Modal Seg.* input is required, we provide visible-part segmentation masks generated by SAM2. *BBox* indicates that the method requires a ground-truth bounding box to define the area for amodal completion. Despite this significant advantage, SDAmodal's performance does not surpass that of many methods lacking this extra information. We also distinguish between models by their *Run Type*. Online methods process video sequentially and are suitable for real-time application, whereas offline methods process the entire clip at once, granting them far more context at the cost of higher latency. A significant performance trade-off exists between modelling paradigms. Video Transformer methods like TCOW offer substantially lower latency, often by orders of magnitude, compared to iterative Video Diffusion approaches such as TABE and diffusion VAS.

Quantitative results are shown in Table 4. A clear trend emerges where video-based methods (e.g., TABE, diffusion-vas, TCOW) generally outperform image-based approaches (pix2gestalt, SDAmodal). This is expected, as video models can leverage temporal context, which is particularly useful for interpolating across a full occlusion. A notable exception is the lower performance of Generative Omnimatte; this result should be contextualised, as our evaluation protocol intentionally

| Method | Inputs | Run Type | mIOU$_{fo}$ | mIOU$_{ffo}$ | mIoU$_{occ}$ |
|---|---|---|---|---|---|
| pix2gestalt | Modal Seg. | Online | 0.411 $_{\pm 0.2167}$ | 0.060 $_{\pm 0.1074}$ | 0.136 $_{\pm 0.1088}$ |
| TCOW | / | Offline | 0.426 $_{\pm 0.2304}$ | 0.179 $_{\pm 0.1907}$ | 0.165 $_{\pm 0.1324}$ |
| SDAmodal | Modal Seg. & GT BBox | Online | 0.526 $_{\pm 0.2433}$ | 0.059 $_{\pm 0.1176}$ | 0.146 $_{\pm 0.1293}$ |
| Generative Omnimatte | Modal Seg. | Offline | 0.416 $_{\pm 0.2194}$ | 0.045 $_{\pm 0.1284}$ | 0.062 $_{\pm 0.1002}$ |
| TABE | Modal Seg. | Offline | 0.501 $_{\pm 0.2768}$ | 0.209 $_{\pm 0.2572}$ | 0.207 $_{\pm 0.1760}$ |
| Diffusion VAS | Modal Seg. | Offline | **0.604** $_{\pm 0.2580}$ | **0.267** $_{\pm 0.2944}$ | **0.262** $_{\pm 0.2087}$ |

Table 4: Quantitative results on complete Real-VAS dataset.

| Method | Real-VAS Containment | | | Real-VAS Occlusion | | |
|---|---|---|---|---|---|---|
| | mIOU$_{fo}$ | mIOU$_{ffo}$ | mIoU$_{occ}$ | mIOU$_{fo}$ | mIOU$_{ffo}$ | mIoU$_{occ}$ |
| pix2gestalt | 0.203 $_{\pm 0.11}$ | 0.016 $_{\pm 0.03}$ | 0.053 $_{\pm 0.04}$ | 0.478 $_{\pm 0.20}$ | 0.096 $_{\pm 0.13}$ | 0.163 $_{\pm 0.11}$ |
| TCOW | 0.325 $_{\pm 0.14}$ | **0.108** $_{\pm 0.14}$ | **0.128** $_{\pm 0.11}$ | 0.458 $_{\pm 0.24}$ | 0.236 $_{\pm 0.21}$ | 0.177 $_{\pm 0.14}$ |
| SDAmodal | 0.335 $_{\pm 0.19}$ | 0.022 $_{\pm 0.04}$ | 0.076 $_{\pm 0.05}$ | 0.587 $_{\pm 0.23}$ | 0.088 $_{\pm 0.15}$ | 0.168 $_{\pm 0.14}$ |
| Generative Omnimatte | 0.250 $_{\pm 0.15}$ | 0.000 $_{\pm 0.00}$ | 0.003 $_{\pm 0.01}$ | 0.470 $_{\pm 0.21}$ | 0.082 $_{\pm 0.16}$ | 0.081 $_{\pm 0.11}$ |
| TABE | 0.225 $_{\pm 0.14}$ | 0.044 $_{\pm 0.06}$ | 0.118 $_{\pm 0.09}$ | 0.590 $_{\pm 0.25}$ | 0.340 $_{\pm 0.28}$ | 0.236 $_{\pm 0.19}$ |
| Diffusion VAS | **0.368** $_{\pm 0.20}$ | 0.066 $_{\pm 0.14}$ | 0.125 $_{\pm 0.13}$ | **0.681** $_{\pm 0.23}$ | **0.428** $_{\pm 0.29}$ | **0.306** $_{\pm 0.21}$ |

Table 5: Quantitative evaluation on Real-VAS, separated by Containment and Occlusion scenarios.

withholds the foreground (occluder) masks that the model is designed to use, thereby highlighting its dependency on that specific input. To further analyse model performance, Table 5 presents results split by our Occlusion and Containment scenarios. We observe a consistent drop in performance for all methods on the Containment tasks when compared to the simpler Occlusion events. This underscores the difficulty of these interactive scenarios and confirms the importance of including them in a comprehensive amodal segmentation benchmark. It is noteworthy, however, that TCOW outperforms all other methods on Containment mIOU$_{ffo}$ and mIoU$_{occ}$, this is likely attributable to its training on synthetic containment-style videos, which demonstrates that performance on these complex physical reasoning tasks is heavily influenced by exposure to similar training data.

# 6 DISCUSSION

In this work, we addressed the critical challenge of creating a large-scale, high-fidelity dataset for video amodal segmentation. We introduced Real-VAS, a novel dataset generated through a sophisticated pipeline that leverages real-world footage to produce pixel-perfect ground-truth annotations. This approach successfully bridges the sim2real gap found in purely synthetic data while bypassing the inaccuracies and high costs associated with manual annotation or 3D reconstruction. A key contribution of our work is the introduction of the Container scenarios, which push beyond the simple pass-by occlusions prevalent in existing datasets. By providing structured, physically-grounded interactions such as On Surface Containment, Articulated Containment, and Mobile Containment, Real-VAS establishes a new and challenging benchmark for evaluating a model's capacity for physical reasoning and object permanence. The inclusion of both Occlusion and Container scenarios is deliberate, as we hypothesise they require different model capabilities. For instance, the structured logic of Containment may be solvable by models that excel at physical or relational reasoning, yet these same models might fail at the visual completion needed for unstructured Occlusion. Conversely, powerful generative models that can hallucinate appearance for Occlusion events may lack the explicit physical constraints required to succeed at Containment - our dataset is designed to spur the development of unified solutions that can solve both. Our pipeline further enhances realism by emulating dynamic camera motion and incorporating secondary effects like cast shadows.

**Limitations and Future Work** While our compositing pipeline is designed to be highly realistic, it cannot capture all worldly phenomena. The ground truth for our container scenarios relies on a "snugly fit" physical constraint, which does not cover all containment possibilities, such as a small object moving loosely inside a large box. Furthermore, while we composite real footage, subtle lighting inconsistencies or contact point artifacts can occasionally occur. Future work could extend our generation pipeline to include controlled real-world camera motion, fluid dynamics, or more complex self-occlusion.

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

# APPENDICES

## A  FURTHER DATASET GENERATION DETAILS

### A.1  SEGMENTATION CLASS LIST

To identify objects that are inherently dynamic, we define a comprehensive set of 'moving object' classes. This set is broadly categorized to cover a wide range of potential motion.

It includes biological entities such as person, animal, bird, dog, cat, fish, horse, and insect. It also features a wide array of vehicles, from terrestrial (car, bus, motorcycle, train) and marine (boat, ship) to aerial (airplane, helicopter, drone). Furthermore, we include mechanical systems and devices known for their movement, like robot, conveyor belt, fan, and clock hands. Finally, the list is rounded out by common recreational and everyday objects whose behavior is often dynamic, including ball, kite, swing, and shopping cart.

### A.2  USE OF COTRACKER3 WITHIN CONTAINMENT LOGIC

For our ground-truth generation in containment scenarios, we found that running CoTracker3 in online mode was significantly more robust than in offline mode. This is because the offline tracker, which processes the entire video at once, was prone to identity switching. In cases where multiple identical containers crossed paths, the offline model would incorrectly assign the track to a different container that later occupied the same space. The online, frame-by-frame approach mitigates this issue by maintaining a continuous track without being influenced by future events.

### A.3  COMBINING MONOCULAR DEPTH WITH SEGMENTATION

In cases where segmentation (SAM2) struggles with fine details, we leverage monocular depth (Depth Anything V2) to refine compositing. By calculating the average depth within the mask, we can exclude regions with greater depth values, effectively isolating the background. This technique substantially improves the realism of our composites. As shown in Figure 5, it allows for visibility through intricate structures, like the gaps between the leaves of a plant, correctly revealing the parts of the person.

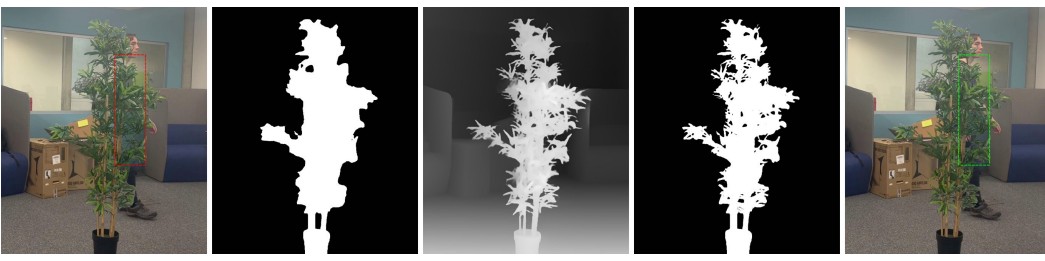

Figure 5: Refining a segmentation mask using monocular depth. From left to right: (a) The initial composite showing the person occluded by the plant (missing upper torso and upper leg). (b) The coarse mask from SAM2. (c) The corresponding depth map. (d) The refined mask after using depth to exclude background pixels. (e) The final, improved composite, where details of the person are correctly revealed through the gaps in the plant.

### A.4  DATASET GENERATION SPEED AND COMPUTATION REQUIREMENTS

Our data generation pipeline was executed on an NVIDIA A40 GPU. While it is not strictly a requirement, a GPU with a large VRAM is beneficial for processing long video sequences, where the computational demands of the SAM2 mask propagation step can be significant.

Quantifying the precise generation speed is challenging due to the highly parallelised and automated nature of the pipeline. In particular, the clip alignment stage, where our system searches for plausible occlusion events, was designed to run overnight on large batches of videos to maximise the number of high-quality candidates discovered.

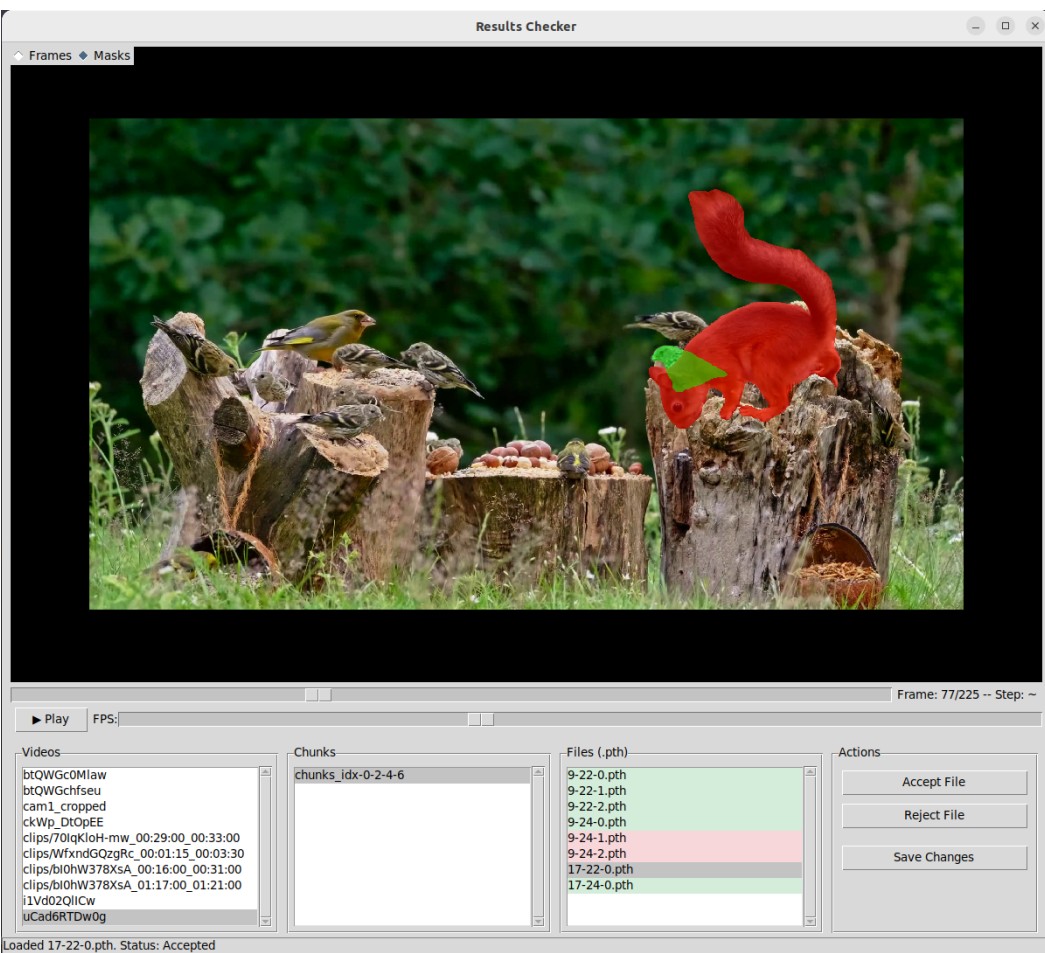

Figure 6: Our simple GUI for the final verification step, where generated clips are quickly accepted or rejected based on their overall quality.

## A.5    MANUAL QUALITY CONTROL VERIFICATION STAGES

While our data generation pipeline is fully automated, we incorporate two quality control checks on the videos processed in the Real-VAS dataset to ensure the final dataset meets a high standard of quality and accuracy required for an evalutation dataset. We have built tools to ensure these QC checks are highly efficient.

First, following the automated mask generation, we perform a rapid curation step. Using a lightweight, purpose-built GUI (Fig. 7), human curators can quickly cycle through masks to correct obvious segmentation errors or add any that were missed. This interface is designed for speed, turning a potentially laborious task into a quick quality check that focuses only on refining clear inaccuracies.

Second, after a full sequence has been composited, it undergoes a final acceptance review. Curators use a simple viewing tool to watch the short clip and give a quick pass/fail judgement based on its overall realism and interest, as shown in Figure 6. This final check is extremely fast and ensures that only the most compelling and accurate sequences are included in the dataset. By designing these lightweight interfaces, our human-in-the-loop stages serve as an efficient quality control mechanism rather than a time-consuming annotation bottleneck.

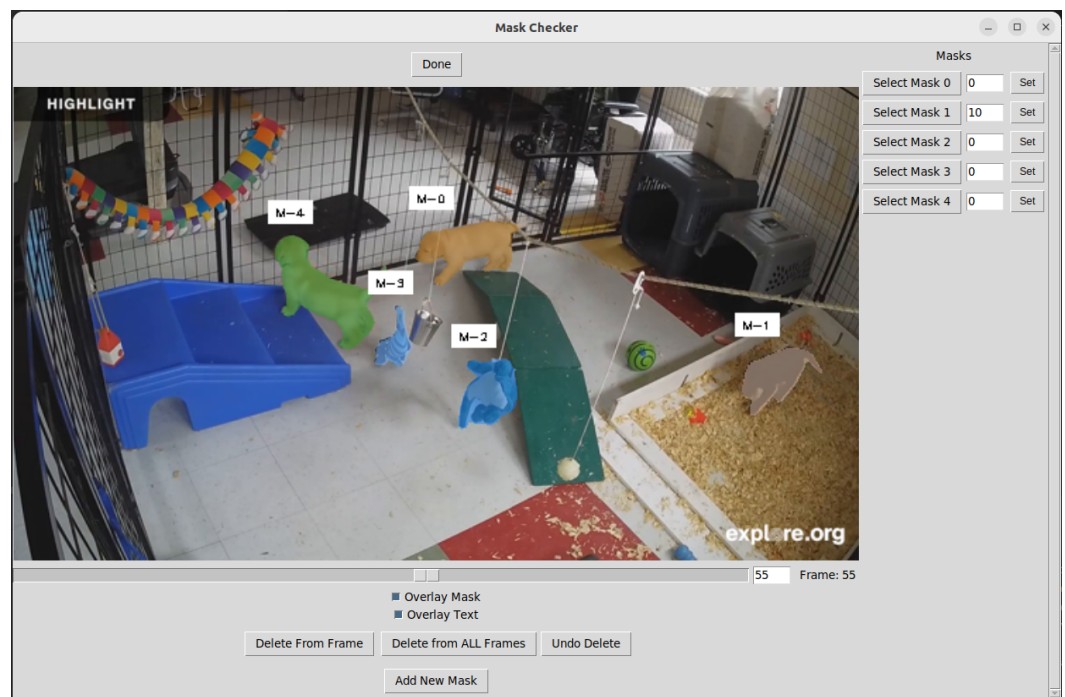

Figure 7: The GUI for the mask verification step. This tool allows users to easily correct inaccuracies by adding, modifying, or removing masks.

## B  ABLATION STUDIES

### B.1  PRECISION OF CONTAINMENT LOGIC

A key challenge for our Containment scenarios is that once an object is hidden, its true position cannot be visually verified. To quantitatively validate the precision of our ground-truth generation logic, we therefore conducted a specialised experiment using videos of transparent containers.

This validation process involves two steps. First, we establish a verifiable ground truth by running a standard segmentation model on the original transparent videos. Since the object is always visible through the container, this provides its true mask for every frame. Second, we digitally render the transparent containers opaque in these same videos and then run our proposed containment logic. This logic takes the object's last visible mask before it becomes hidden and uses a point tracker on the (now opaque) container to predict the object's subsequent amodal masks.

Across all our container scenarios, comparing our logic's predicted masks against the verifiable ground truth yields a mean IoU of 97.5%. While this score is very high, we note that IoU is highly sensitive to minor pixel shifts, and the segmentation masks used as a reference are not perfectly noise-free. Nevertheless, this result demonstrates that our containment logic is highly precise. We showcase the experimental setup for each scenario in Fig. 8.

### B.2  DATASET SCALE

To validate Real-VAS as a robust evaluation benchmark, we performed a scaling analysis. We measured the performance of the top performing baseline methods on progressively larger random subsets of our test data, from 10% to 100%. The results (Fig. 9) demonstrate two key properties.

First, the changing scores in the initial subsets (e.g., from 10% to 70%) highlight the dataset's *diversity*, showing that different parts of the dataset contain unique challenges that test various model capabilities. More importantly, as the evaluation size increases towards its full scale (from 80% to 100%), the performance scores for all models begin to stabilise and the curves flatten.

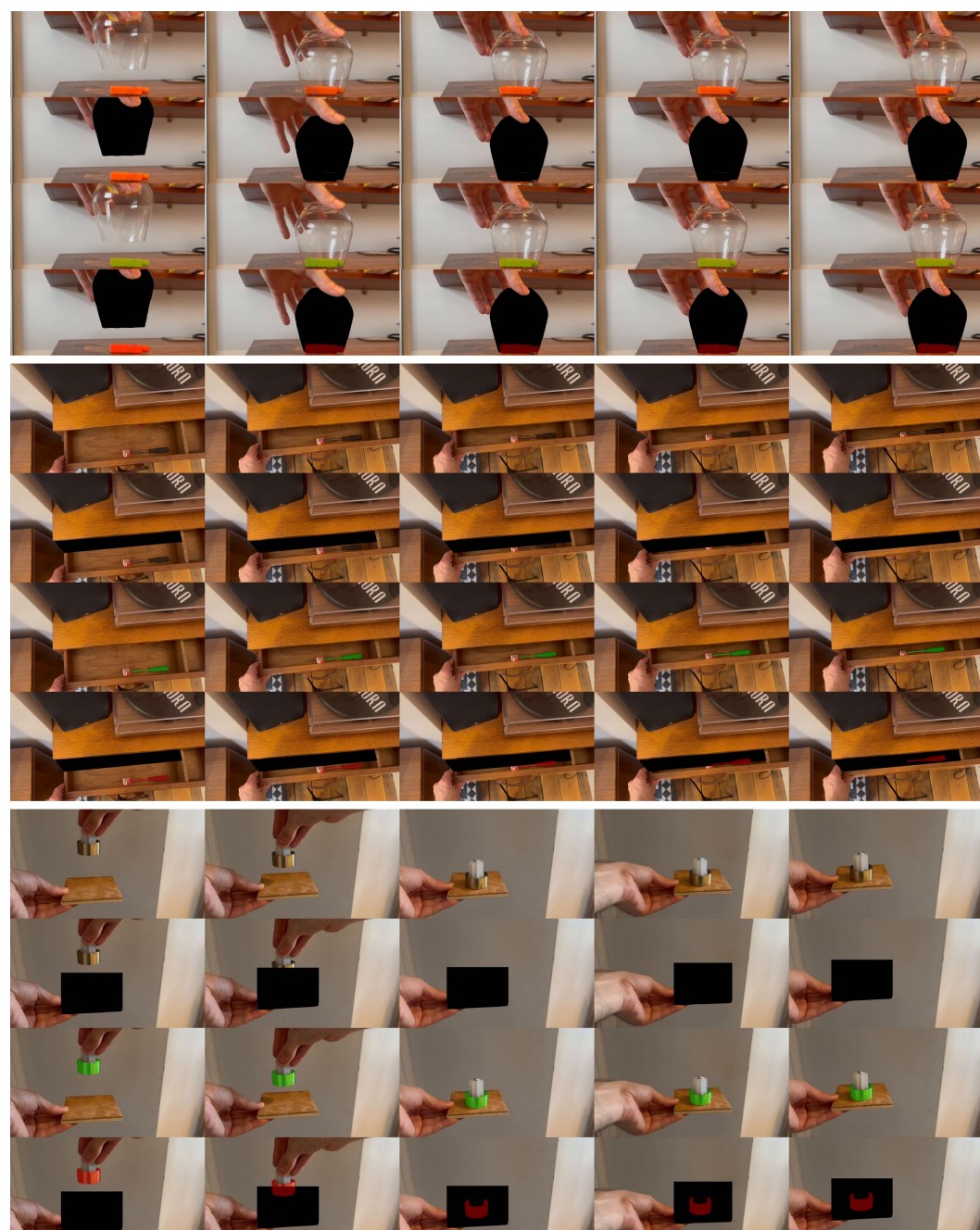

Figure 8: Validation of our containment logic across three key scenarios: On-Surface, Articulated, and Mobile Containment. For each example, the rows display (from top to bottom): the original frame, the synthetically applied container occlusion, the ground truth mask, and the final mask predicted by our logic. Our approach is highly accurate, achieving a 97.5% Intersection over Union (IoU) with the ground truth masks.

This convergence is critical. It indicates that while an evaluation on a small subset might be sensitive to the specific scenes chosen, an evaluation on the full dataset provides a stable and reliable measure of a model's true performance. The fact that the scores stabilise demonstrates that Real-VAS is *sufficiently large and comprehensive* to serve as a conclusive benchmark.

In Table 6 we showcase some additional statistics of our Real-VAS dataset supplementing the comparisons in 1. These metrics highlight the dataset's diversity, showing a wide variation in clip

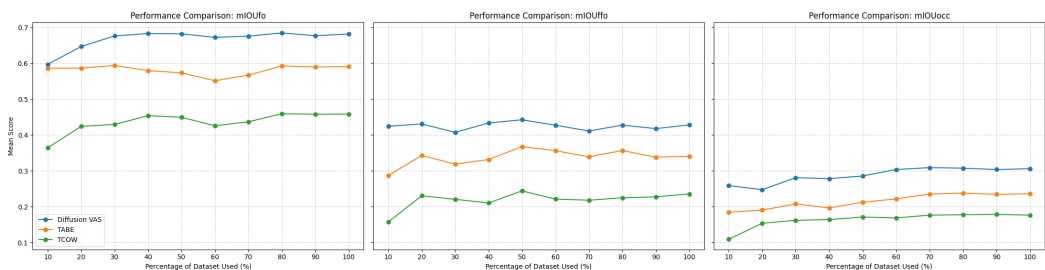

Figure 9: Scaling analysis of model performance on the Real-VAS dataset. The plots show the mean score for three different mIoU metrics as the percentage of the evaluation data is increased from 10% to 100%. The general convergence of the curves validates the dataset's scale as a stable and comprehensive benchmark.

lengths, frame rates and resolutions. Furthermore, Table 7 demonstrate good coverage across our Occlusion and Container scenarios.

| Clip Length (frames) | | | Frames Per Second | | | Resolution | | |
|---|---|---|---|---|---|---|---|---|
| Min | Max | Avg | Min | Max | Avg | Min | Max | Mode |
| 14 | 329 | 54 | 10 | 60 | 16 | $480 \times 640$ | $1080 \times 1920$ | $1080 \times 1920$ |

Table 6: Further statistics of our Real-VAS Dataset.

| Num. Occlusion Clips | Num. Containment Clips | | | |
|---|---|---|---|---|
| | Total | On Surface | Articulated | Mobile |
| 300 | 100 | 35 | 36 | 29 |

Table 7: The distribution of clips between our Occlusion and Container categories. The number of container scenarios is smaller by design, reflecting the inherent difficulty of sourcing the specific, high-quality physical interactions required for this category. In contrast, simple occlusion events are far more common in in-the-wild footage, allowing for a greater volume and variety.

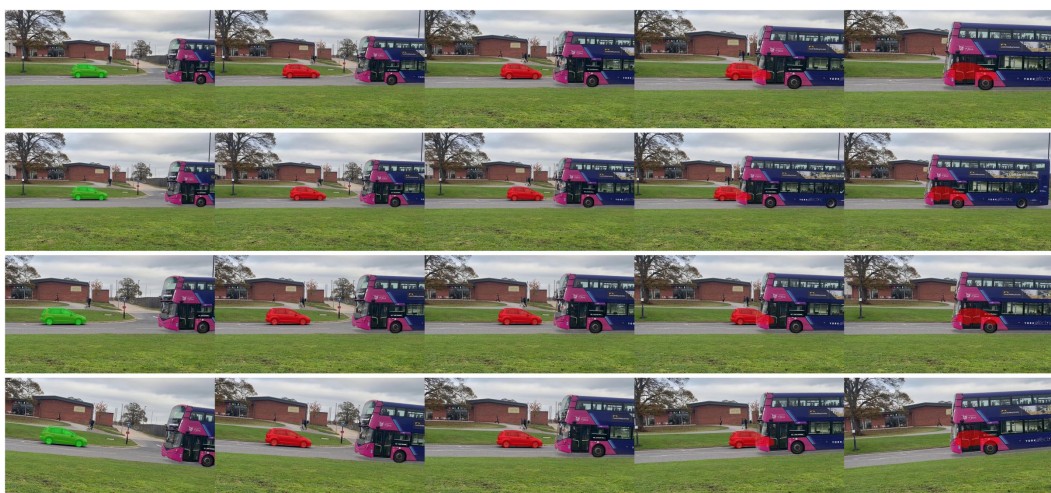

Figure 10: A selection of example outputs from our camera emulation module. The figure displays four profiles from our larger library of predefined motions, which are applied to static videos to enhance realism and introduce challenging, dynamic viewpoints to the dataset.

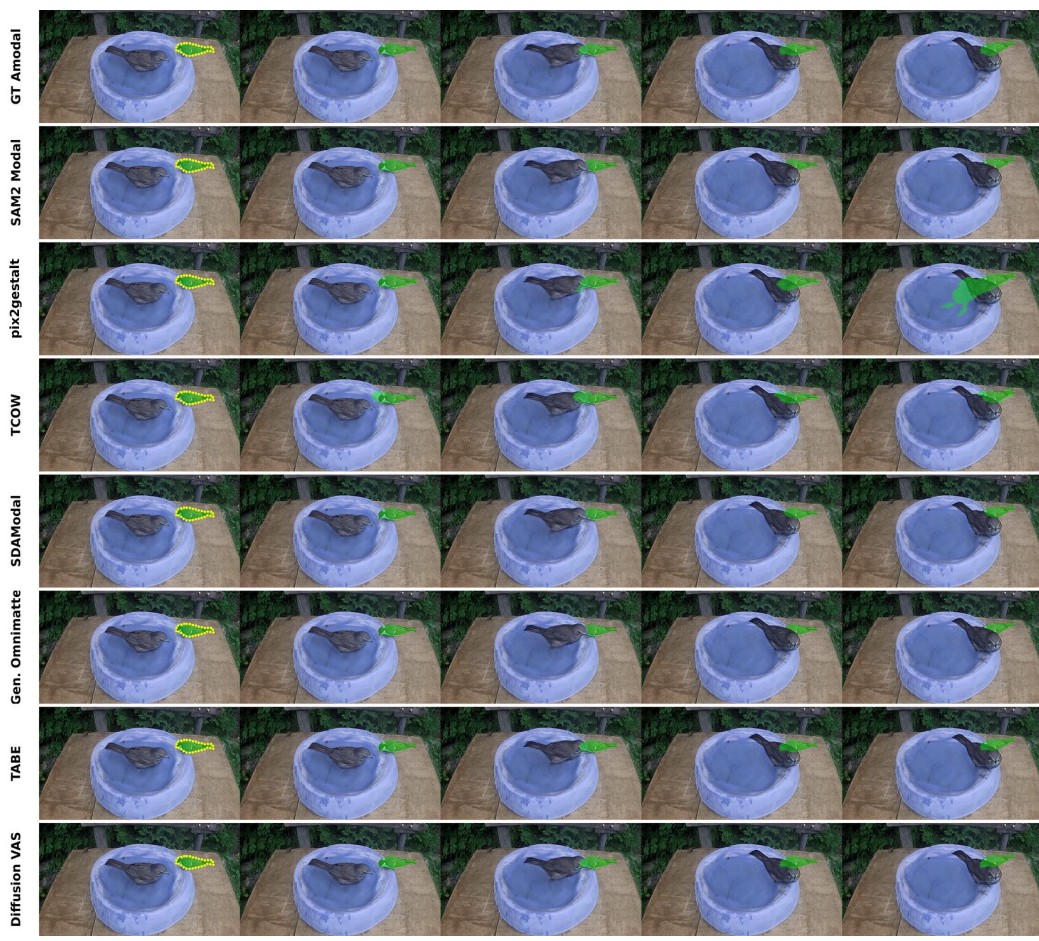

Figure 11: Qualitative outputs from each method - Part 1

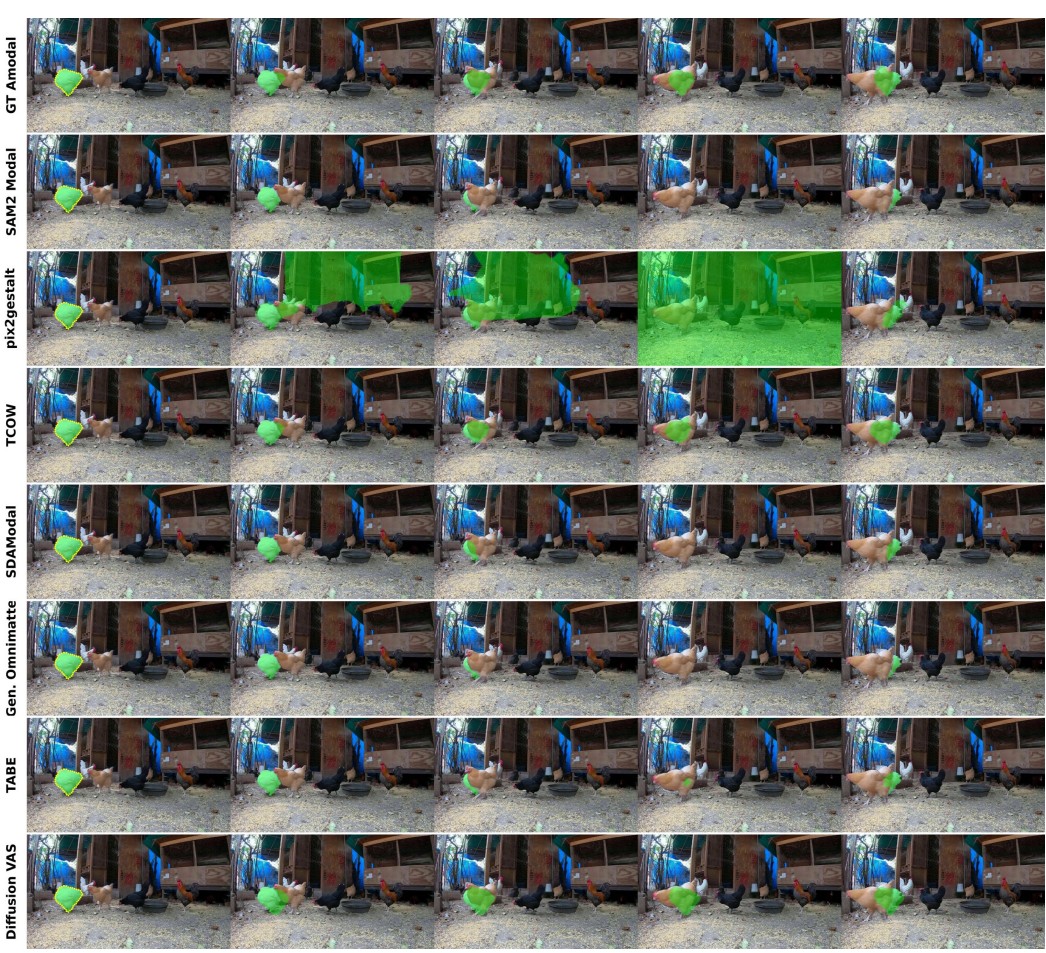

Figure 12: Qualitative outputs from each method - Part 2

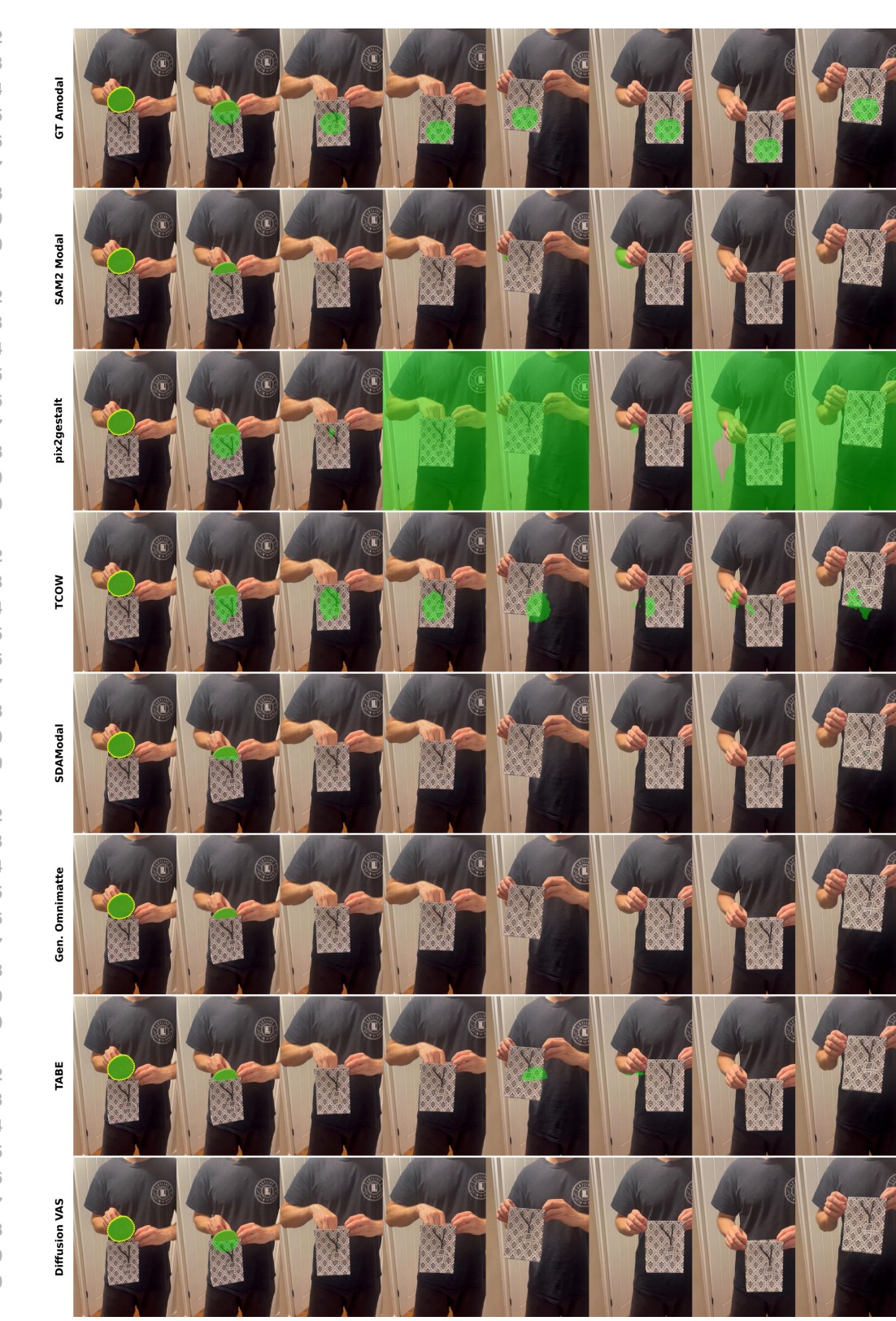

Figure 13: Qualitative outputs from each method - Part 3

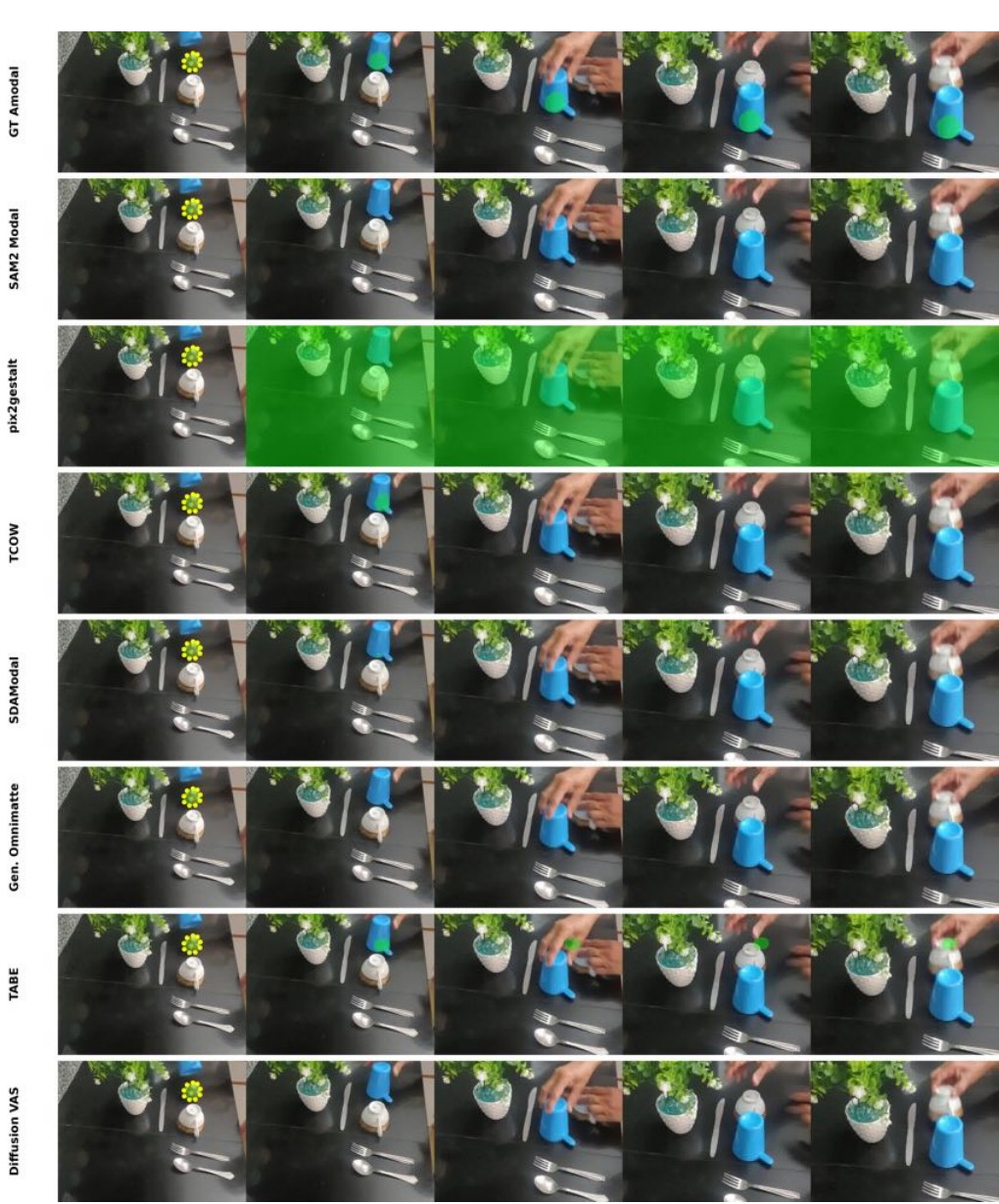

Figure 14: Qualitative outputs from each method - Part 4

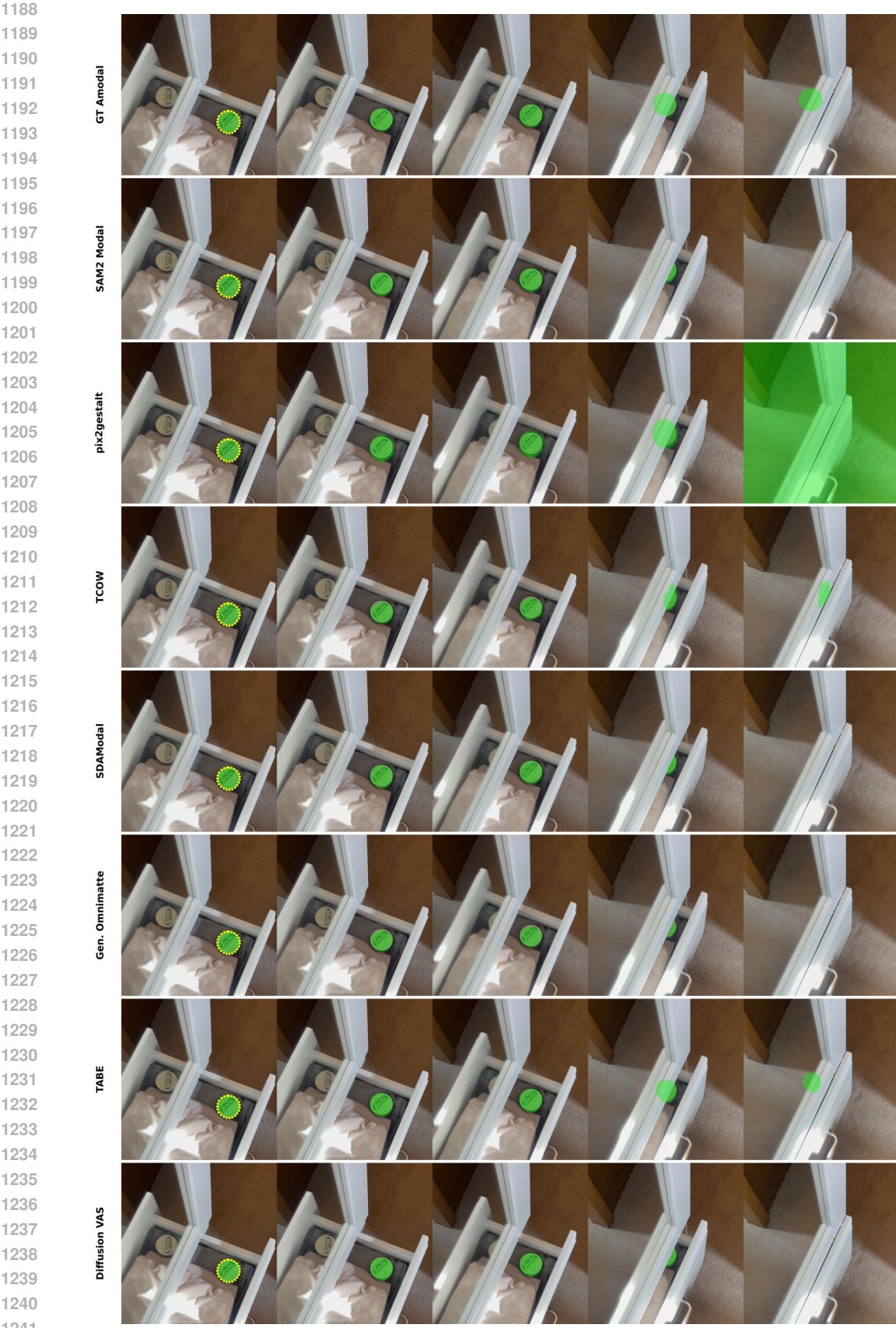

Figure 15: Qualitative outputs from each method - Part 5

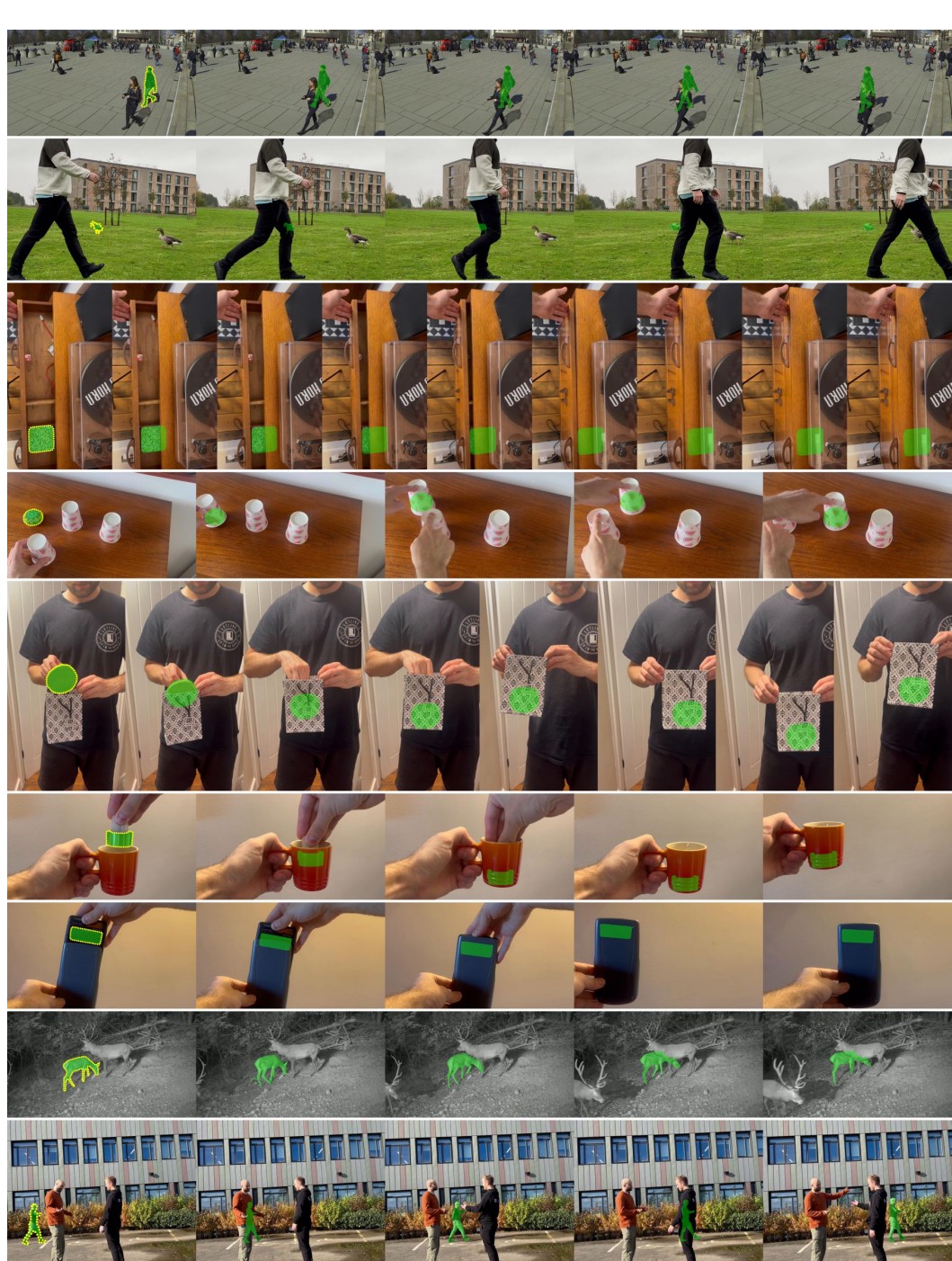

Figure 16: Further examples from the Real-VAS dataset. The yellow dashed line indicates the initial query mask, followed by the ground-truth amodal segmentation frames.

