# OpenReview forum: "Real-VAS: a Realworld Video Amodal Segmentation dataset"
_ICLR.cc/2026/Conference — Submitted to ICLR 2026_

### Official Review · Reviewer_pb5w · 2025-10-18

**Soundness:** 2
**Presentation:** 2
**Contribution:** 3
**Rating:** 6
**Confidence:** 3

**Summary:**

The paper presents Real-VAS, a large-scale zero-shot evaluation dataset for video amodal segmentation, aiming to solve the key limitations of existing datasets, such as the sim2real gap in synthetic data and inaccurate human-estimated annotations in real-world data. It develops two automated data generation pipelines to produce pixel-perfect ground truth without relying on expensive 3D reconstruction or manual annotation. The dataset includes novel Containment scenarios and adopts three metrics (mIoU_fo, mIoU_ffo, mIoU_occ) to evaluate SOTA models, demonstrating its effectiveness as a challenging VAS benchmark.

**Strengths:**

1. The dataset generation method combines real-world video clips with automated ground-truth generation. It bridges the sim2real gap of synthetic datasets and avoids the inaccuracy and high cost of manual annotation for real-world data .

2. The paper proposes mIoU_fo and mIoU_ffo to solve the misleading issue of standard mIoU, making VAS model evaluation more accurate.

3. The Real-VAS dataset contributes to VAS research.

**Weaknesses:**

1. Limited Containment scenario coverage: The Containment pipeline relies on a "snugly fit" physical constraint, restricting scenario diversity.

2. Incomplete related work: The paper omits an earlier paper [1] that focuses on video amodal segmentation and was published prior to all the methods cited in its related work section.
In the related work section, the authors only discuss their contributions related to synthetic datasets and fail to mention this earlier paper in the method part. This undermines the comprehensiveness of the literature review.

3. Data generation heavily depends on tools like SAM2 (for segmentation) and CoTracker3 (for tracking); errors in these tools may propagate to the dataset’s ground truth, reducing its reliability

[1] Self-supervised amodal video object segmentation.

**Questions:**

1. In the Occlusion pipeline of Real-VAS, a mean motion score S (calculated via CoTracker3) and a predefined threshold $\tau$ are used to filter dynamic occluders and occludees. How was the specific value of threshold $\tau$ determined, and were ablation studies conducted to verify its rationality in ensuring both dataset quality and scenario diversity?

2. What technical challenges need to be addressed to extend the Containment pipeline to such loose interaction scenarios, and are there any preliminary solutions?

---

> ### Author Response · Authors · 2025-11-17
>
> *Limited Containment scenario coverage: The Containment pipeline relies on a "snugly fit" physical constraint, restricting scenario diversity*
>
> We agree with the reviewer that our current containment pipeline is constrained by a "snugly fit" assumption, a point we acknowledge as a limitation in our paper. However, we must emphasise two key points. First, our work introduces the first benchmark for this type of complex, physically-grounded containment scenario, moving significantly beyond the simple "pass-by" events in existing datasets. Second, our evaluation demonstrates that this "limited" scenario is already exceptionally challenging for current state-of-the-art methods.As shown in Table 5, all models experience a substantial performance drop on the Containment split. Notably, the best-performing model for this task, TCOW, only achieves an $mIoU_{occ}$ of 0.128. This result underscores that even this foundational version of the problem is far from being solved. While expanding diversity is an important next step, our current dataset provides a critical and demonstrably difficult new benchmark for the field.
>
> *Incomplete related work: The paper omits an earlier paper [1] that focuses on video amodal segmentation and was published prior to all the methods cited in its related work section.*
>
> This was an unintentional oversight. We agree that this paper is a foundational contribution to video amodal segmentation. In the revised manuscript, we will, of course, add this citation and discuss its contributions within related work
>
> *Data generation heavily depends on tools like SAM2 (for segmentation) and CoTracker3 (for tracking); errors in these tools may propagate to the dataset’s ground truth, reducing its reliability*
>
> We agree this is a valid concern, as no automated tool is infallible. This is precisely why we do not rely solely on the outputs of SAM2 and CoTracker3. As detailed in Appendix A.5 of our paper, we implemented a multi-stage manual quality control pipeline to ensure the reliability of our ground truth. This essential human-in-the-loop process is designed to specifically identify and correct potential segmentation or tracking failures from the automated tools, thereby guaranteeing the final accuracy of the benchmark
>
> *In the Occlusion pipeline of Real-VAS, a mean motion score S (calculated via CoTracker3) and a predefined threshold τ are used to filter dynamic occluders and occludees. How was the specific value of threshold τ determined, and were ablation studies conducted to verify its rationality in ensuring both dataset quality and scenario diversity?*
>
> The specific value of $\tau$ was determined via qualitative empirical analysis. We inspected the correlation between the calculated motion score $\mathcal{S}$ and the actual visual motion of objects across a sample dataset.We chose a strict threshold to act as a high-quality filter, ensuring that the dataset predominantly features objects with significant, challenging motion. It is important to note that this automated threshold is not the only quality control mechanism. As described in Appendix A.5, we employ a rigorous manual verification stage. This human oversight ensures that the final dataset maintains high quality and appropriate diversity, correcting for any potential limitations of the automated filtering parameter.
>
> *What technical challenges need to be addressed to extend the Containment pipeline to such loose interaction scenarios, and are there any preliminary solutions?*
>
> Our "snugly fit" constraint is a core part of our methodology, as it allows us to deterministically infer the object's amodal mask by tracking the container. In a "loose interaction" (e.g., a small object in a large box), this link is broken, and the object's true position is unknown. As for potential solutions: Several avenues exist, each with trade-offs.
>
> 1.  Multi-Camera Capture: One could use a complex capture setup, such as a bird's-eye-view camera, to track the object's true position inside the container.
>
> 2.  Digital Occlusion: Another approach, similar to our validation method in Appendix B.1, would be to film with transparent containers and then digitally render them opaque.
>
> 3.  Synthetic Occluders: A third option is to synthetically add a realistically modelled physics-based occluder to a real scene.However, we argue that solutions (2) and (3) risk compromising the real-world fidelity that is a core goal of our work.Current Difficulty: Most importantly, we must stress that even our current, constrained benchmark is far from being solved. As our results in Table 5 show, all SOTA methods perform exceptionally poorly on the Containment split. The best-performing model only achieves an $mIoU_{occ}$ of 0.128. This demonstrates that the field must first solve this foundational problem before increasing the challenge with loose interactions. We are excited to see the community build upon our work to solve both.

---

> > ### Comment · Reviewer_pb5w · 2025-11-27
> >
> > Thanks for your response and for addressing my question. I keep my score.

---

### Official Review · Reviewer_CAUF · 2025-10-27

**Soundness:** 2
**Presentation:** 2
**Contribution:** 2
**Rating:** 2
**Confidence:** 4

**Summary:**

This paper introduces the Real Video Amodal Segmentation (Real-VAS) dataset, designed to balance data collection scale with real-world fidelity. It also presents two data generation pipelines: an occlusion pipeline, which blends two video clips and leverages Language Segment-Anything and SAM 2 to construct occlusion scenarios, and a containment pipeline, which uses SAM 2 and CoTracker3 to track objects before and after occlusion. Evaluations on Real-VAS show that video-based approaches outperform image-based ones, and that training on similar data could benefit performance.

**Strengths:**

1. The integration of multiple classic and state-of-the-art computer vision algorithms and models to enhance real-world fidelity provides a promising direction for addressing realistic data scarcity in the amodal segmentation domain.
2. The dataset and code will be open-sourced.

**Weaknesses:**

This paper should be rejected due to two key weaknesses:
1. **The claim of "pixel-perfect amodal ground truth for real-world video" is too weak**
   - The occlusion pipeline uses soft alpha blending, which makes the occluder semi-transparent. This is unrealistic and not truly *amodal*, as amodal segmentation requires full occlusion.
   - The containment pipeline, which applies the container's transformation (tracked by CoTracker3) to the occludee, does not produce pixel-perfect ground truth either.
2. **The evaluation is limited**
   - The evaluation experiments provide no novel insights. Simply showing that video models perform well on video domains, or that models trained on containment data perform well on containment domains, offers little value to the community.
   - Some methods perform very poorly on Real-VAS, but the paper does not analyze why or how to improve them. Even for a dataset paper, there should be evidence showing how this dataset can inspire new approaches and advance amodal segmentation by revealing new insights.
   - Several standard models, such as [PCNet-M (CVPR 2020)](https://xiaohangzhan.github.io/projects/deocclusion/) and [AISFormer (BMVC 2022)](https://uark-aicv.github.io/AISFormer/), are missing from the evaluation. Including them would better demonstrate the dataset's reliability as a zero-shot amodal segmentation benchmark.

Things to improve the paper that did not impact the score:
1. **Figure 1:** Use the same example throughout the occlusion pipeline for better clarity.
2. **Table 1:** It would be helpful to include all related datasets (e.g., TAO-Amodal) and highlight key differences such as *bbox vs mask* to provide a clearer comparison supporting the claim of being large-scale.
3. **Table 5:**
   - Ensure consistent capitalization between "mIOU" and "mIoU."
   - Add citations for each method.
   - Clearly separate image-based and video-based methods.

**Questions:**

One key question the authors need to address:
* The arXiv paper ["Track Anything Behind Everything: Zero-shot Amodal Video Object Segmentation"](https://arxiv.org/abs/2411.19210) was explicitly **cited** in the *Related Work: Amodal Segmentation Methods* section and referred to multiple times throughout the paper for its method **TABE**, but its dataset, **TABE-51**, was intentionally omitted.
* For example:
  * This paper's **Figure 4** example and that paper's **Figure 7** (bottom-left) example are almost identical - two people talking in front of a building.
  * This paper's **Figure 1** alpha-blend example and that paper's **Figure 3** example are almost identical - a person walking through a door.
  * This paper's **Table 2** is exactly the same as that paper's **Table 2**.

What is the difference between this work and TABE-51? Is this work built on top of it?

Other questions:
- More details about the dataset: e.g., how many samples are from in-house sources and how many are web-crawled? What is the size of the containment and occlusion subsets, respectively?
- As a data generation pipeline designed for scalability, what are the inference speed and cost to produce a single data point?

**Details Of Ethics Concerns:**

Currently raised the concerns in the Questions section, as this is an arXiv only paper explicitly cited in the submission.
* The arXiv paper ["Track Anything Behind Everything: Zero-shot Amodal Video Object Segmentation"](https://arxiv.org/abs/2411.19210
) was explicitly **cited** in the submission, but its dataset, TABE-51, was intentionally omitted.

---

> ### Author Response · Authors · 2025-11-17
>
> *amodal segmentation requires full occlusion*
>
> We thank you for your comment, but there appears to be a significant misunderstanding of our compositing pipeline. The "soft alpha blending" applies only to boundary pixels to simulate sub-pixel precision and motion blur. The occluder's interior remains fully opaque ($\alpha=1.0$), preventing the occluded object from being seen. We use Generative Omnimatte precisely to capture these realistic edge effects and shadows, which creates a more realistic composite than hard-edged overlays, as shown in Figure 3. We believe the confusion may arise from our qualitative figures (e.g., Figures 2, 11-15). In these figures, the ground-truth amodal mask is intentionally overlaid with transparency for visualisation purposes only, so the reader can assess the occluded shape. In the actual dataset videos provided to the models, the occluder is opaque, and the occludee is completely hidden, making the task truly amodal. Within supplementary materials we provide a zip file with some examples of the dataset, which shows the opacity of the occluder, the only thing visible is the amodal mask which we have additionally overlaid for debug visualisation.
>
> *Containment pipeline does not produce pixel-perfect ground truth*
>
> We respectfully disagree with this assessment. We quantitatively validated the precision of our containment logic, as this was a key challenge in our pipeline. As detailed in Appendix B.1, we conducted a specialised experiment to verify our "pixel-perfect" claim. The core challenge is that a hidden object's true position cannot be visually verified. To overcome this, we captured video of scenarios using transparent containers and generated a verifiable ground truth by segmenting the (always visible) object. Before digitally rendering these containers opaque and then running our proposed containment logic (tracking the opaque container with CoTracker3 and applying its transformation). When comparing our logic's predicted amodal masks against the verifiable ground truth, our method achieved a mean IoU of 97.5\%. As shown in Figure 8, this result confirms that our containment logic is highly precise. The minor discrepancy is due to IoU's high sensitivity to single-pixel shifts and minute noise in the reference masks.
>
> *Details about the dataset*
>
> As described in Section 4.1, our dataset is built from two sources to ensure diversity: in-house capture and web-scraped videos. Of the 400 videos, 252 are from web-crawled sources and 148 are from our in-house collection. Regarding the subset sizes, this information is detailed in Table 7. The dataset is composed of 300 Occlusion videos and 100 Containment videos.  We will ensure the in-house vs. web-crawled breakdown is stated clearly in the final paper.
>
> *Models missing from evaluation*
>
> PCNet-M: We evaluated this method as requested. It achieved very low scores ($mIoU_{fo}=0.208$, ($mIoU_{ffo}=0.001$, $mIoU_{occ}=0.005$), further confirming the challenging nature of the Real-VAS benchmark.
>
> AISFormer: This model was excluded as it is incompatible with our task definition. Real-VAS evaluates zero-shot video amodal segmentation, which we define as a class-agnostic paradigm for segmenting novel objects via a query mask. AISFormer is a class-specific model limited to only outputting masks from predefined training categories, making it unsuitable for this zero-shot benchmark
>
> *This work and TABE-51*
>
> We appreciate the reviewer noting the similarity to [Hudson \& Smith 2024]. We'd like to respectfully note that per ICLR's review guidelines, arxiv preprints are not considered prior published work for the purposes of evaluating novelty, and reviewers are instructed to ignore arxiv papers that they believe might be early versions of work under review. We cite [Hudson \& Smith 2024] because their amodal segmentation method serves as one of the baselines in our experimental evaluation.
>
> *inference speed and cost*
>
> We appreciate this question. As detailed in Appendix A.4, our generation pipeline is designed for high-throughput batch processing on an NVIDIA A40 GPU. Cost and Speed Estimate: For our Occlusion pipeline, we observe an average production rate of approximately 1 minute per data point (based on a yield of ~61 candidates in an hour on a video instance). The primary "cost" is therefore the GPU time required for the search and the subsequent storage. Caveats on Scalability: It is important to note that this speed is not uniform.  As noted in the paper, the generation speed is highly dependent on the input video content. The Containment pipeline is naturally more computationally expensive per second of footage due to the complex logic required to track containers and transform object masks, though it constitutes a smaller portion of the total dataset. We have prioritised precision and realism over raw speed. There is significant room for future optimisation, particularly in the candidate search phase

---

> ### Comment · Reviewer_CAUF · 2025-11-18
> **Regarding TABE model and TABE-51 dataset**
>
> Thanks for addressing the comment!
> The major point is if we cited the paper [Hudson & Smith 2024], it cannot be ignored now.
> According to the [reviewer guideline](https://iclr.cc/Conferences/2026/ReviewerGuide):
> > While authors are not required to compare to contemporaneous work or unpublished arxiv papers, they are strongly encouraged to cite such related work if they are aware of it.
>
> Then apparently we are aware of it. And how can we be only aware of its model but not its dataset? Please help to address this.

---

> > ### Author Response · Authors · 2025-11-20
> >
> > We are awaiting advice from the AC on how best to preserve anonymity while further replying to your comment. We will respond as soon as we have been given more guidance.

---

> > > ### Comment · Reviewer_CAUF · 2025-11-20
> > >
> > > Thank you for following up on this. Prior to writing the original review, we were also awaiting for AC's response regarding this corner case for clarification. Agreed we could wait till the official response out.
> > >
> > > If the arXiv paper had not been cited, we would have ignored it. However, since it is **explicitly** cited yet a substantial portion of the work is intentionally omitted in the paper, we believe it is appropriate to flag this at least for completeness. Thank you for your understanding.

---

> > > > ### Author Response · Authors · 2025-11-25
> > > >
> > > > We're still waiting on the AC's advice. Subject to this issue being resolved, do any of your other concerns still need discussion or have they been dealt with in our rebuttal?

---

### Official Review · Reviewer_xFwj · 2025-10-31

**Soundness:** 4
**Presentation:** 4
**Contribution:** 4
**Rating:** 6
**Confidence:** 3

**Summary:**

This paper introduces Real-VAS, a real-world video amodal segmentation dataset for zero-shot evaluation. It generates pixel-perfect ground truth using automated compositing and container-tracking pipelines, covering dynamic occlusion and containment scenarios. Real-VAS combines realism and precision, providing a challenging benchmark for testing physical reasoning and occlusion understanding.

**Strengths:**

The paper demonstrates strong originality by introducing a novel real-world video amodal segmentation dataset (Real-VAS) that overcomes the long-standing gap between synthetic precision and real-world realism. Its quality is supported by a well-designed automated pipeline that produces pixel-perfect annotations without manual estimation. The clarity of presentation, with detailed figures and methodological descriptions, makes the contribution easy to follow. Finally, its significance lies in establishing a challenging benchmark that can drive progress in zero-shot amodal segmentation and physical reasoning in vision models.

**Weaknesses:**

Although Real-VAS introduces an important real-world benchmark, its dataset scale remains limited. The relatively small number of videos may restrict the statistical robustness of model evaluation and the diversity of object interactions. To strengthen its impact, the authors could expand the dataset with more varied scenes, objects, and motion patterns, or release additional data splits to enhance coverage and generalization testing.

**Questions:**

Have the authors compared model performance on Real-VAS with results on other real or synthetic amodal segmentation datasets? In other words, if a model performs well on Real-VAS, does that success transfer to other datasets, or is Real-VAS capturing distinct challenges? Such comparisons would help clarify how well the benchmark aligns with or extends existing datasets in terms of generalization and real-world difficulty.

---

> ### Author Response · Authors · 2025-11-17
>
> *Dataset scale remains limited*
> We thank you for raising this important point about scale. We would like to address this in three parts:
>
> Scale in Context: The term "large-scale" is relative. While 400 videos may seem modest, this corresponds to 21,436 annotated frames. More importantly, as we state in Section 4.4, Real-VAS provides almost four times more frames than the next largest real-world benchmark that offers accurate, pixel-level amodal annotations. While synthetic datasets are larger, they suffer from the well-known sim2real gap. Our contribution is specifically a large-scale real-world dataset, which addresses a critical gap in the field.
>
> Containment Data: We agree that the Containment data (100 clips ) is, by design, more "cumbersome" to produce and thus smaller than the Occlusion set (300 clips ). This was a deliberate methodological choice. Our priority was to introduce a novel, high-quality challenge that was entirely missing from the field, favouring scenario diversity over repetitive volume. Our 100 clips are distributed across three distinct physical interaction types (On Surface, Articulated, and Mobile) to ensure models are tested on a range of reasoning tasks, rather than hundreds of repeating, simplistic instances
>
> Sufficient Scale for Benchmarking: The most critical point is not just the absolute size, but whether the dataset is large and diverse enough to provide a stable evaluation. We directly address this in our Appendix B.2 scaling analysis. As shown in Figure 9, performance on small subsets (10-70%) is volatile, highlighting the dataset's diversity. However, as the evaluation size approaches 100%, "the performance scores for all models begin to stabilise and the curves flatten". This convergence "demonstrates that Real-VAS is sufficiently large and comprehensive to serve as a conclusive benchmark".
>
> *Have the authors compared model performance on Real-VAS with results on other real or synthetic amodal segmentation datasets...*
>
> We agree that such comparisons are crucial. We analysed the DiffusionVAS method, which provides a useful case study as it was evaluated on both synthetic data and our Real-VAS. This model was benchmarked on SAILVOS, a large-scale synthetic occlusion dataset made from the GTA5 game, where it achieved an mIoUocc of 0.55. In sharp contrast, when we evaluate the same model on our Real-VAS Occlusion split, its performance drops significantly to 0.31 mIoUocc. This substantial gap underscores the increased difficulty and distinct challenges presented by real-world video compared to synthetic data. Furthermore, the model's performance deteriorates to just 0.13 mIoUocc on our Real-VAS Containment split. This result is particularly revealing, as it highlights that our containment data introduces a novel and difficult challenge that existing benchmarks do not capture, and for which current models are not optimised. This confirms that Real-VAS effectively tests challenges not present in existing datasets.
>
> *Evaluation of the constructed data as a training set*
>
> We are not proposing Real-VAS as a training set and have not conducted any experiments to use it as such. The area of Video Amodal Segmentation currently lacks a meaningful benchmark and this is the first goal that Real-VAS aims to fill: provide a test set that can honestly assess the state-of-the-field and compare methods. By presenting our scaleable dataset methodology and implementation of this to the community, a scaled-up version of this dataset could certainly be built for training in future.
>
> *For the REAL-VAS CONTAINMENT part...*
>
> We are not finetuning CoTracker3; we use it strictly as an offline tool to generate ground truth. The concern that models might just "fit CoTracker3" overlooks the task. A "direct combination" of VOS and CoTracker3 requires a handcrafted heuristic to tell the system when an object is contained and what acts as the container. The purpose of Real-VAS is to evaluate if models can learn to perform this complex reasoning autonomously from visual context, rather than relying on the hard-coded logic we used to create the dataset. You have struck at the very heart of the rationale as to why we created a dataset with such diverse containment and occlusion tasks. While it is possible to construct heuristics for specific cases—for instance, hard-coding a rule to "follow the cup" in an On Surface scenario, such an approach is inherently fragile. Not only would this be prone to identity switches during complex motion, but it would also fail catastrophically when applied to other scenarios, such as Mobile or Articulated Containment or the occlusion's category. The purpose of Real-VAS is to move beyond brittle, hand-crafted logic and encourage less task specific training. By presenting a wide variety of physical interactions, we compel models to learn a robust, unified representation of object permanence that can autonomously solve all scenarios without task-specific engineering.

---

### Official Review · Reviewer_CGx8 · 2025-11-04

**Soundness:** 3
**Presentation:** 3
**Contribution:** 2
**Rating:** 4
**Confidence:** 4

**Summary:**

This paper explores the construction of a real-world amodal video dataset (implied as Real-VAS). To build this dataset, the authors first identify potential occlusion events from static camera videos, then leverage physical law-based models as constraints, overlaying the cropped mask sequence of one object onto another to realize the construction of amodal videos. The paper provides a detailed introduction to the dataset construction method and conducts ablation studies to validate its effectiveness. This work not only enriches the data resources for the field of amodal video segmentation but also offers a feasible technical framework for real-world amodal dataset construction, holding certain significance for the development of the field.

**Strengths:**

1. The authors constructed amodal videos through clip overlapping from the perspective of ensuring the laws of real physical motion, which features high reliability. This approach may provide some insights to the development of this field and robotics.
2. It is very reasonable to use models such as Depth Anything V2 and Generative Omnimatte to ensure physical plausibility.
3. The writing is good and the paper is easy to follow.

**Weaknesses:**

1. For the REAL-VAS CONTAINMENT part, the motion of objects is mainly controlled by CoTracker3, which raises a serious concern: for containment data, is the trained model actually fitting CoTracker3? What advantages does this have compared with directly combining CoTracker3 with ordinary VOS/VIS models? Directly combining CoTracker3 seems to be more robust.
2. All videos are sourced from static cameras, which limits the diversity of videos, even though simulated camera motion has been added.
3. The proposed Real-VAS is described as a "large-scale" dataset, but the total number of videos is only 400, which is not a large scale. In addition, for containment-type data, fixed cameras and scenes are required for shooting, and the cumbersome production process seems unsuitable for scaling up.
4. There is a lack of evaluation of the constructed data as a training set.
5. The analysis in Table 3 is very valuable, but please add a brief introduction to TABE and Diffusion VAS.

**Questions:**

How is the motion of objects inside the container modeled? For example, in the third row of Figure 2, when an object is put into a paper bag, how is the speed at which the object falls to the bottom of the paper bag determined? And as the paper bag moves, will the object move inside the container? This is more reflective of the laws of physics but seems to have not been explored.

---

> ### Author Response · Authors · 2025-11-17
>
> *How is motion of objects inside the container modeled?*
>
> Excellent question. The motion is not determined by a physics-based simulation of falling. Instead, we infer the object's movement by tracking the actuator (the hand) that places it inside the container. In the Figure 2 example, we use our point tracker (CoTracker3) on a visible reference point of the hand, such as a knuckle. During the insertion—as the object becomes occluded by the bag—we maintain a fixed hand-pose assumption. The tracked motion of the knuckle is then directly applied to the object's last-known mask to determine its speed and final "settled" position within the bag. This multi-stage process, which we refer to as "simulating the object settling", is what allows us to generate ground truth for this initial interaction. Once the hand is removed, the logic switches to tracking the container itself. We will clarify this mechanism in the final paper, likely with a supplementary figure.
>
> *And as the paper bag moves, will the object move inside the container?*
>
> This is a critical point and relates directly to the "snugly fit" physical constraint we detail in the paper. The object does not move independently inside the container. Our ground-truth generation relies on the object and container moving as a single rigid unit once containment is complete. To ensure this, we sometimes modified the container. In the paper bag example, the bag was lightly padded on the interior, making it a "snugly fit" so the object was fixed in place. This modification was only necessary for a few scenarios to broaden the diversity of objects and containers. In many other cases, such as several examples in Figure 16, the object-container pair naturally met this snug-fit requirement without alteration. We thank the reviewer for raising this and will add a supplementary figure or note to the final manuscript to make this methodological detail clearer.
>
> *All videos are sourced from static cameras...*
>
> We thank you for this point. While our Containment pipeline can handle moving cameras, our Occlusion pipeline does require a static camera for its compositing logic. We openly acknowledge this as a limitation in Section 6. However, our key finding is that even with this simplification, the benchmark is far from solved. Our emulated camera motion (Section 4.3) is not a trivial addition; it is a validated pipeline that introduces challenging dynamic viewpoints. Most importantly, the performance of all SOTA models on our dataset is exceptionally poor, as detailed in Table 5. The best $mIoU_{occ}$ on our novel Containment split is a mere 0.128. This demonstrates that Real-VAS is already a sufficiently difficult challenge for the field. Solving this is a critical prerequisite before adding the significant complexity of uncontrolled, real-world camera motion, which, as we note, is an important direction for future work.
>
> *The analysis in Table 3 is very valuable...*
>
> We will add descriptions for TABE and Diffusion VAS to Section 5 as suggested.
>
> *The proposed Real-VAS is described as a "large-scale" dataset...*
>
> We thank you for raising this important point about scale. We would like to address this in three parts:
>
> 1. Scale in Context: The term "large-scale" is relative. While 400 videos may seem modest, this corresponds to 21,436 annotated frames. More importantly, as we state in Section 4.4, Real-VAS provides almost four times more frames than the next largest real-world benchmark that offers accurate, pixel-level amodal annotations. While synthetic datasets are larger, they suffer from the well-known sim2real gap. Our contribution is specifically a large-scale real-world dataset, which addresses a critical gap in the field.
>
> 2. Containment Data: We agree that the Containment data (100 clips ) is, by design, more "cumbersome" to produce and thus smaller than the Occlusion set (300 clips ). This was a deliberate methodological choice. Our priority was to introduce a novel, high-quality challenge that was entirely missing from the field, favouring scenario diversity over repetitive volume. Our 100 clips are distributed across three distinct physical interaction types (On Surface, Articulated, and Mobile)  to ensure models are tested on a range of reasoning tasks, rather than hundreds of repeating, simplistic instances
>
> 3. Sufficient Scale for Benchmarking: The most critical point is not just the absolute size, but whether the dataset is large and diverse enough to provide a stable evaluation. We directly address this in our Appendix B.2 scaling analysis. As shown in Figure 9, performance on small subsets (10-70\%) is volatile, highlighting the dataset's diversity. However, as the evaluation size approaches 100\%, "the performance scores for all models begin to stabilise and the curves flatten". This convergence "demonstrates that Real-VAS is sufficiently large and comprehensive to serve as a conclusive benchmark".

---

### Meta-Review · Area_Chair_xK62 · 2026-01-07

**Summary:**

Reviewers praised the novel pipeline for generating pixel-perfect ground truth in real-world scenarios but expressed valid concerns regarding the dataset's limited scale (400 videos) and restricted containment physics. Importantly, as pointed out by Reviewer CAUF, the authors explicitly cite a previous arXiv version of their submission. I recommend a desk rejection because it breaches double-blind anonymity.

**Reviewer Concerns:**

The rebuttal successfully clarified ground truth reliability via validation experiments and justified the "snug fit" constraint by highlighting the poor performance of current SOTA models on the benchmark. However, the procedural dispute regarding the citation of the authors' prior arXiv work remains outstanding, while concerns about the dataset's small scale persist for some reviewers.

**Reviewer Scores:**

Reviewers pb5w and xFwj likely maintain their positive scores (6) following satisfactory technical clarifications, while CGx8 (4) likely remains borderline due to scale limitations. Reviewer CAUF's score (2) hinges entirely on the administrative ruling regarding the self-citation and anonymity issue.

---

### Decision · Program_Chairs · 2026-01-26

Reject